# Nucleotide– and Mal3-dependent changes in fission yeast microtubules suggest a structural plasticity view of dynamics

Ottilie von Loeffelholz[1,3], Neil A. Venables[2,4], Douglas Robert Drummond[2,5], Miho Katsuki[2,6], Robert Cross[2] & Carolyn A. Moores[1]

Using cryo-electron microscopy, we characterize the architecture of microtubules assembled from *Schizosaccharomyces pombe* tubulin, in the presence and absence of their regulatory partner Mal3. Cryo-electron tomography reveals that microtubules assembled from *S. pombe* tubulin have predominantly B-lattice interprotofilament contacts, with protofilaments skewed around the microtubule axis. Copolymerization with Mal3 favors 13 protofilament microtubules with reduced protofilament skew, indicating that Mal3 adjusts interprotofilament interfaces. A 4.6-Å resolution structure of microtubule-bound Mal3 shows that Mal3 makes a distinctive footprint on the *S. pombe* microtubule lattice and that unlike mammalian microtubules, *S. pombe* microtubules do not show the longitudinal lattice compaction associated with EB protein binding and GTP hydrolysis. Our results firmly support a structural plasticity view of microtubule dynamics in which microtubule lattice conformation is sensitive to a variety of effectors and differently so for different tubulins.

[1] Institute of Structural and Molecular Biology, Birkbeck College, London WC1E 7HX, UK. [2] Division of Biomedical Cell Biology, Warwick Medical School, CV4 7AL Coventry, UK. [3] Present address: Centre for Integrative Biology, Department of Integrated Structural Biology, Institute of Genetics and of Molecular and Cellular Biology, 1 rue Laurent Fries, Illkirch, France. [4] Present address: CRUK Manchester Institute, The University of Manchester, Wilmslow Road, Manchester M20 4BX, UK. [5] Present address: Centre for Promotion of International Education and Research, Faculty of Agriculture, Kyushu University, Fukuoka 812-8581, Japan. [6] Present address: Department of Earth System Science, Faculty of Science, Fukuoka University, Fukuoka 814-0180, Japan. Correspondence and requests for materials should be addressed to C.A.M. (email: c.moores@mail.cryst.bbk.ac.uk)

The dynamic properties of microtubules (MTs) are a crucial facet of their contribution to cell function. αβ-tubulin heterodimers polymerize head to tail to form polar protofilaments (PFs), while lateral inter-PF contacts complete the hollow MT wall[1,2]. These lateral contacts can be homotypic (so-called B-lattice) or—where α-tubulin contacts β-tubulin—can form a so-called seam or A-lattice contact[3]. While both α- and β-tubulin bind GTP[4,5], the intrinsic dynamic instability of MTs is driven by the GTPase cycle of β-tubulin. GTP-bound αβ-tubulin stimulates MT nucleation and growth, while polymerization stimulates GTP hydrolysis in β-tubulin, producing MTs that are intrinsically unstable. These properties thus enable coupling between the tubulin GTPase and MT polymerization/depolymerization dynamics[2], although the structural basis for this coupling continues to be investigated.

MT dynamics and architectures are tightly controlled in vivo, such that the dominant architecture is 13-PF B-lattice MTs with a single A-lattice seam[6–9]. However, cryo-EM studies of in vitro-polymerized MTs reveal the heterogeneity of the resulting architectures, reflecting the potential plasticity of intersubunit contacts within the MT lattice[10]. In vitro studies have also shed light on the influence of bound nucleotide on the MT lattice, revealing longitudinal structural compaction in response to the tubulin GTPase[11]. More recently, near-atomic resolution MT reconstructions have allowed visualization of the structural impact of the β-tubulin-bound nucleotide on MTs, revealing the local conformational adjustments that accompany switching between the so-called extended (GMPCPP/GTP-like) and compacted (GDP.Pi/GDP-like) MT states[12].

These basal nucleotide-dependent conformational dynamics of tubulin determine the structural properties of MTs built from pure tubulin. In addition, the conformation of tubulin can be switched, modified, and even overridden by a variety of MT interactors, and the properties that emerge greatly extend the repertoire of MT structure, function, and dynamics[13]. The mechanisms by which specific MAPs can facilitate MT nucleation, stabilization, or depolymerization are of intrinsic interest, and can furthermore shed light on the underlying conformational properties of tubulin.

End-binding proteins (EBs) are of particular interest in this context. EBs form dynamic clusters or "comets" at the ends of growing, but not shrinking, MTs[14]. Comet formation by EBs on growing MTs can be recapitulated in vitro, suggesting that these proteins have the intrinsic ability to set or sense the underlying structure of the growing MT tip[15–17]. Several studies have shown a strong tendency of EBs to drive MTs to assemble with 13 PFs, with comet intensity consistent with near-stoichiometric EB occupancy of the comet MT lattice[18–20]. A number of mechanisms have been proposed to explain these properties, including binding to MT seams[21], assembly of A-lattice MTs[19,22], and sensitivity to the underlying MT GTPase state[16,17,20,23]. More recent work, including higher-resolution cryo-EM studies, shows EB binding to the B-lattice, between neighboring PFs at the corners of four tubulin dimers within the MT lattice[20,23] or in MT sheets[24]. Such a binding site would be predicted to be sensitive to nucleotide-dependent changes within the MT lattice. Reconstructions of EB–MTs polymerized in the presence of the slowly hydrolyzable GTP analog GTPγS reveal that the EB-preferred/stabilized conformation of MTs is compacted (presumed to be post GTP hydrolysis[23]). It is, however, unclear how the two apparently diametrically opposed views of the binding specificity of EB proteins, namely A-lattice vs. B-lattice binding, can be reconciled. One possibility is that EBs genuinely have two or more binding modes. At some level, this is already evident, since EBs bind to the GDP lattice and the GTP (and/or GDP.Pi) cap, but with different affinities.

In addition to nucleotide-dependent conformational changes, tubulin from mammalian brain contains multiple α- and β-tubulin isoforms that are also subject to numerous post-translational modifications[25]. Development of biochemical tools for recombinant preparation of tubulin and of purification of tubulin from nonbrain sources has begun to allow investigation of the contributions that specific tubulin isotypes and PTMs make to MT structure and dynamics[26,27]. Purification of biochemically useful amounts of genomically encoded tubulin from Schizosaccharomyces pombe[28] has enabled in vitro studies of this tubulin[19,22,29,30], providing a valuable complement to in vivo studies of S. pombe MT dynamics, e.g.,[31–35]. Given the different dynamics and organization of MTs in yeast and mammals, it is of great interest to investigate the properties of MTs polymerized from this tubulin and to probe the consequences for S. pombe MAP binding specifically.

To illuminate the structural–dynamical aspects of EB binding to MTs, we have studied MTs polymerized in vitro from purified native tubulin from S. pombe (Sp_tub) together with the S. pombe EB protein Mal3 using cryo-EM. We have also correlated our results with solution studies of Mal3-GFP binding to dynamic MTs of both Sp_tub and mammalian brain tubulin (Mam_tub) using TIRF microscopy. Our data identify key differences in the behavior of Sp_tub compared to that of Mam_tub. Further, our results test the generality of the currently proposed relationship[23] between nucleotide state, subunit compaction, and PF skew in the nucleotide-control mechanism of MT assembly.

## Results

**MT architecture of S. pombe tubulin in vitro.** To investigate polymerization of Sp_tub in vitro, MTs prepared in a range of conditions were examined using cryo-EM. Specifically, we used cryo-electron tomography (cryo-ET) to visualize the 3D architecture of Sp_tub MTs in an unbiased way. MTs were decorated with a kinesin motor domain after polymerization to emphasize the underlying tubulin dimer repeat and thereby facilitate characterization of MT architecture (which did not itself perturb the architecture, Table 1, Supplementary Table 1). For each MT that underwent 3D reconstruction, the moiré repeat in 2D—which arises from the superposition of the PFs in the 3D MT wall—was also measured (Fig. 1a)[10]. The number of PFs from which the MT was built was determined using multivariate statistical analysis[36] of transverse 2D sections extracted along the MT length from the 3D tomogram (Fig. 1b, Table 1).

Dynamic Sp_tub MTs polymerized with GTP (GTP-Sp_tub MTs) are mainly built from 13 PFs (Table 1), as was found earlier for mammalian MTs, e.g.,[10,37]. However, in contrast to mammalian 13-PF MTs—which have extremely long moiré repeats (>1 μm) in which the PFs are essentially straight—GTP-Sp_tub 13-PF MTs display a range of moiré repeats with a median length of ~500 nm. This means that on average, GTP-Sp_tub MTs exhibit a greater PF skew than mammalian MTs. By contrast, Sp_tub MTs stabilized by GMPCPP, a nonhydrolyzable GTP analog, exhibit an extremely short median moiré repeat of ~100 nm, indicative of an even larger PF skew. These GMPCPP MTs are mainly built from 12 and 13 PFs (Fig. 1b,c, Table 1). For comparison, GMPCPP-Mam-tub MTs polymerized under the same conditions yield characteristic 14 PF populations with a moiré repeat of ~500 nm, again corresponding to a more shallow PF twist[11,38]. Overall, these observations point to distinctly different polymerization properties of Sp_tub compared with Mam_tub[39], and are consistent with the PFs in Sp_tub MTs being markedly more skewed within the lattice[19].

**Modulation by Mal3 of *S. pombe* MT architecture in vitro**. Polymerization of Sp_tub in the presence of monomeric Mal3 (1-143, Mal3-143) with either GTP or GMPCPP produced MT populations with higher PF numbers. However, while the majority of Sp_tub MTs have 13 PFs in both the presence and absence of Mal3-143, Mal3-143 strongly disfavors 11 and 12 PF MTs (Fig. 1c, Table 1), broadly consistent with previous reports that coassembly of tubulin with EB family members favors 13-PF MTs[18–20]. With Mal3-143, both GTP and GMPCPP Sp_tub MT populations exhibit a range of moiré repeats indicative of some PF skew, including some MTs with repeats >1000 nm. The PF skew in the presence of Mal3-143 is thus less than that which was seen in its absence, showing that the presence of Mal3-143 during polymerization can influence the architecture of the resulting Sp_tub MTs (Fig. 1c, Table 1) by restricting the PF number and relative PF skew within the lattice.

We also used direct observation of the pattern of kinesin binding on individual MTs to visualize the underlying MT lattice arrangement by cryo-ET. The pattern of kinesin motor domain fiducials was visible in most cases and only a B-lattice-type arrangement was observed (Fig. 1d).

In summary, the polymerization behavior of Sp_tub, with a prevalence of skewed PF MTs in all observed conditions, differs from that observed for Mam_tub. However, the MT architecture parameters extracted from our data sets (Table 1, Supplementary Table 1) are consistent with the range of symmetries built primarily from B-lattice contacts previously described by the lattice accommodation model of MT formation[40]. Although we did not directly observe an A-lattice PF arrangement in our data, A-lattice seams are predicted to be present in the Sp_tub MTs given their architectures[6–9]. Copolymerization of Sp_tub with Mal3-143 tended to converge the MT architecture to a 13-PF arrangement with a slight PF skew, regardless of the bound nucleotide, suggesting that the Mal3-MT interaction is independent of the nucleotide bound to β-tubulin. Our data therefore indicate that coassembly with Mal3-143 adjusts inter-PF interfaces, tending to converge both the shear and spacing between neighboring PFs to a restricted range of values.

**Near-atomic resolution structure of Mal3-bound *S. pombe* MTs**. Bound Mal3 enhances the difference between α- and β-tubulin in the predominantly 13-PF Sp_tub MTs polymerized in the presence of GTP and Mal3-143 (Supplementary Fig. 1a). Using cryo-EM and single-particle averaging approaches[41], we determined the structure of these Mal3-bound GTP-Sp_tub 13-

PF B-lattice MTs with a single A-lattice seam (Fig. 2a, without bound kinesin motor domain) to an overall resolution of 4.6 Å (Supplementary Fig. 1b; see for example Fig. 2b), allowing generation of pseudoatomic models of *S. pombe* α-tubulin1 and β-tubulin, as well as the CH domain of Mal3. Overall, this reconstruction shows the Mal3-143 CH domain bound in-between four αβ-tubulin dimers except at the seam (Fig. 2a), as described previously for EB family proteins[20,23].

This reconstruction allows us to describe the near-atomic structure of polymerized Sp_tub. The overall tubulin fold is conserved within Sp_tub monomers compared to mammalian tubulin, as expected, given the 77% overall sequence identity between *S. pombe* α-tubulin compared to mammalian TUBA1B (Fig. 3a, Supplementary Fig. 2), and 78% identity between *S. pombe* β-tubulin compared to mammalian TUBB1B (Fig. 3a, Supplementary Fig. 3). Consistent with this, structural alignment of individual monomers yields small RMSDs of 1.06 Å for α-tubulin and of 1.26 Å for β-tubulin (Sp_tub vs. Mam_tub, PDB: 3JAR).

When the Sp_tub dimer is superposed by structural alignment on β-tubulin (Fig. 3b,c) larger deviations from the mammalian dimer structure manifest themselves (average RMSD = 1.52 Å). Thus, when the overall configuration of the dimer is considered, the Sp_tub dimer appears to be intrinsically slightly tilted in the context of the lattice compared to Mam_tub. The relatively modest sequence variations between *S. pombe* and mammalian tubulin sequences are distributed throughout both α- and β-tubulin (Fig. 3a), and are not straightforwardly linked to the observed structural differences. However, these structural comparisons suggest that the overall effect of the *S. pombe* tubulin sequences is to render the fundamental Sp_tub building block a slightly different shape compared to Mam_tub, potentially contributing to differences in polymerization behavior including PF skewing.

**S. pombe PF structure reveals an extended conformation**. The average helical repeat distance for the Sp_tub heterodimer measured from the layer line spacing of the MT Fourier transforms was ~83 Å independent of the conditions of MT growth: GTP-Sp_tub = 83.26 +/− 0.01 Å (mean and s.d., n = 82 split into three groups); GMPCPP-Sp_tub = 83.29 +/− 0.11 Å (mean and s.d., n = 94 split into three groups); and Mal3-143 + GTP-Sp_tub = 82.91 +/− 0.02 Å (mean and s.d., n = 316 split into three groups). The difference in repeat length between GTP and GMPCPP Sp_tub MTs is not statistically significant (p = 0.02 (t-test)). The small

**Table 1 Effect of polymerization conditions on Sp_tub MT architecture**

| *S. pombe* tubulin polymerization conditions[a] | | PF number | moiré repeat (nm) (± ~2 nm) | MT number (%) |
|---|---|---|---|---|
| GTP | −Mal3 (n = 25) | 12 | 140–224 | 2 (8%) |
| | " | 13 | 210– >1000 | 21 (84%) |
| | " | 14 | 505–624 | 2 (8%) |
| | + Mal3 (n = 10) | 13 | 400– >1000 | 8 (80%) |
| | " | 14 | >1000 | 1 (10%) |
| | " | 15 | >1000 | 1 (10%) |
| GMPCPP | −Mal3 (n = 30) | 11 | 100 | 1 (3%) |
| | " | 12 | 124–156 | 12 (40%) |
| | " | 13 | 90–144 | 15 (50%) |
| | " | 14 | 90–100 | 2 (7%) |
| | + Mal3 (n = 19) | 12 | >1000 | 2 (11%) |
| | " | 13 | 400– >1000 | 8 (42%) |
| | " | 14 | 140– >1000 | 7 (37%) |
| | " | 15 | 260 | 2 (11%) |

MT architecture parameters were determined by analysis of each cryo-ET 3D volume, and were found to be consistent with the lattice accommodation model proposed by[39,40]
[a]MTs were polymerized as described in the Methods section and KMD was subsequently added prior to cryo-EM sample preparation, apart from the dynamic + GTP MTs, where no KMD was added

difference in the axial subunit spacing of GTP-Sp_tub MTs ± Mal3 is statistically significant ($p < 0.0001$ ($t$-test)), but is very small compared to the equivalent difference seen for mammalian MTs: EB3 + Mam_tub = 81.44 +/− 0.00 Å (mean and s.d., PDB 3JAR[23]); GMPCPP-Mam_tub ~83 Å (3J6G[12]). Thus, the lattice compaction accompanying EB binding/GTP hydrolysis is much smaller in Sp_tub MTs than in Mam_tub MTs.

To understand the consequences of the *S. pombe*-specific properties of the tubulin building block in the context of the MT polymer, we compared the Sp_tub PF structure with that from mammalian MTs. We first compared an overlay of the Mal3-143 + Sp_tub PF with the EB + Mam_tub PF by structural alignment on β-tubulin (as in Fig. 3b). This reveals a large difference between the structures at the ends of the overlaid PFs because the Mal3-143 + Sp_tub PFs are overall more extended than those in the EB + Mam_tub MTs (Fig. 4a), consistent with the measured tubulin dimer repeat distance. Thus, in comparison to the previously reported compaction of EB + Mam_tub MTs, Mal3-143 + Sp_tub adopts a more extended conformation. A similar comparison of Mal3-143 + Sp_tub with Taxol + Mam_tub PFs—previously reported in an extended conformation[12]—reinforces this conclusion (Fig. 4b).

The overlay of the extended Mal3-143 + Sp_tub and Taxol + Mam_tub PFs also shows that there are other structural differences between these polymers besides length, due to the PF skew intrinsic to the Sp_tub MTs (Fig. 4b). Our single-particle calculations show that the average angular relationship between adjacent MT segments boxed along each Mal3-143 + GTP-Sp_tub MT is ~−0.4° +/− 0.3 (median +/− s.d.) relative to the MT axis. According to MT lattice accommodation calculations[39], this corresponds to a PF skew yielding a moiré repeat length of ~700 nm, which is consistent with the range of moiré repeats measured in the cryo-ET data (Fig. 1c, Table 1) and observed directly in the 3D structures of these MTs independently visualized without averaging using cryo-electron tomography (Supplementary Fig. 4). Mal3-143 + GMPCPP-Sp_tub MTs similarly have a PF skew of ~−0.4°, which is reflected in a similar moiré repeat length (see Fig. 1a,c). Intriguingly, this skew angle is similar to that observed for EB + GTPγS-Mam_tub MTs[13,23]. In summary, the axial spacing between subunits in 13-PF Sp_tub is extended compared to mammalian MTs and PF axes are intrinsically skewed. This behavior contrasts with that of 13-PF mammalian MTs, in which skew has only so far been described in the presence of EB[23]. However, despite the difference in lattice spacing, both EB-bound Mam_tub MTs and Mal3-bound Sp_tub MTs exhibit a slight PF skew. EB-driven convergence of PF skew may thus reflect a shared recognition mechanism on different populations of MTs.

The resolution of our reconstruction is not sufficient to unambiguously identify the nucleotide bound at the N-site of α-tubulin (Fig. 4c, presumably GTP, Supplementary Fig. 5a) and the E-site of β-tubulin at the longitudinal dimer interface (Fig. 4d, Supplementary Fig. 5b). However, docking of the EB + Mam_tub E-site structure—in which hydrolysis has occurred—into the

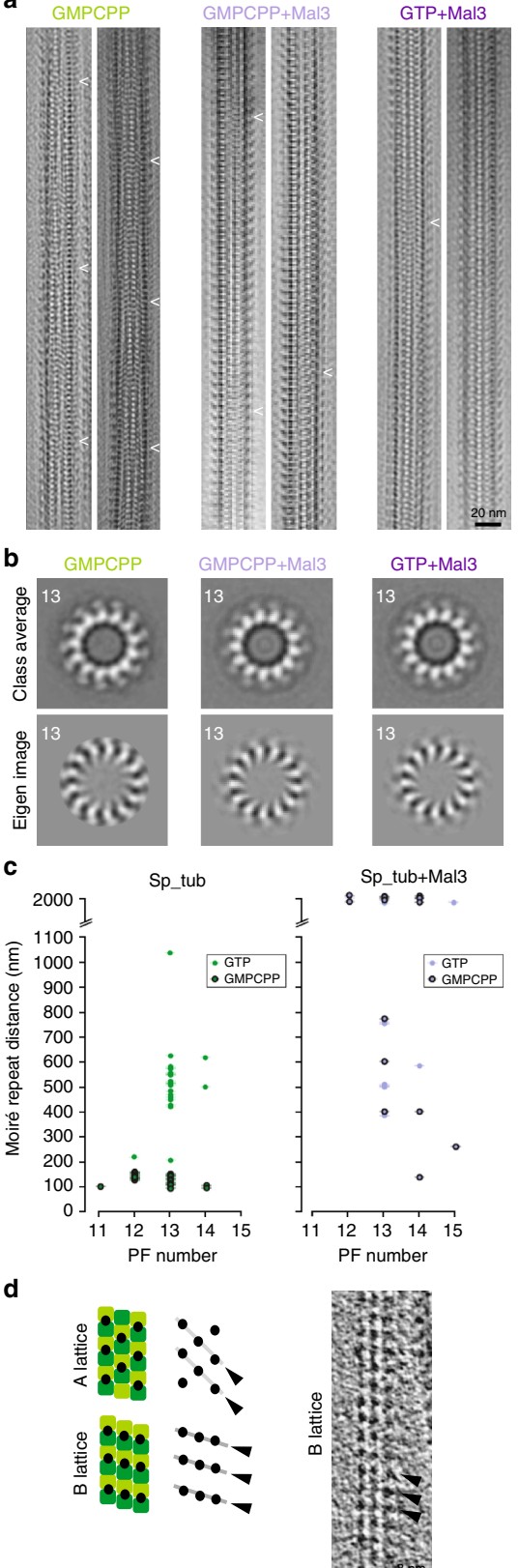

**Fig. 1** Sp_tub MT polymerization architecture and modulation by Mal3. **a** Fourier-filtered cryo-EM MT images of *S. pombe* tubulin polymerized in different conditions as indicated, with a kinesin motor domain added to facilitate characterization of MT architecture. Arrowheads indicate the underlying moiré pattern of each MT. Scale bar = 20 nm. **b** Example class averages and eigen images of 13-PF Sp_tub MTs (among the other PF architectures present) derived from tomograms using multivariate statistical classification. Views presented for the given polymerization conditions in **a**, **b** are not the same MT. **c** Relationship between PF number and moiré repeat length according to the polymerization conditions of Sp_tub. **d** Top, MT A/B-lattices (α-tubulin, dark green, β-tubulin light green), depicted schematically in the presence of KMDs (black circles); bottom, longitudinal section from a tomogram (an example from the GMPCPP + Mal3 data set is shown) allowing visualization of a B-lattice-like KMD arrangement. Scale bar = 8 nm

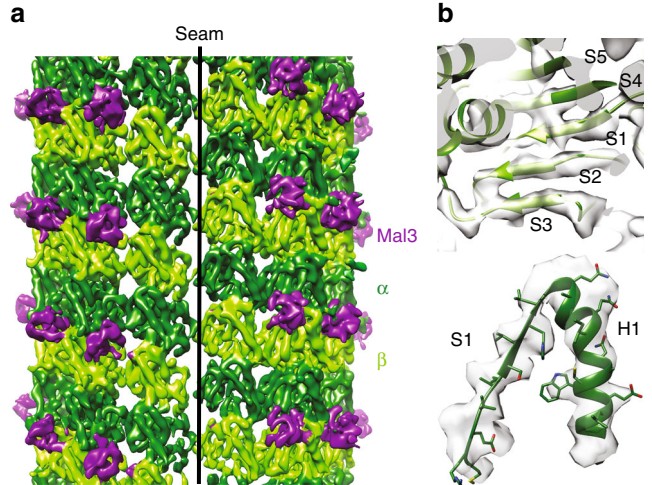

**Fig. 2** Near-atomic resolution reconstruction of Mal3-bound Sp_MT. **a** C1 reconstruction (FSC0.5 = 9.5 Å; 0.143 = 7.5 Å) of Sp-tub GTP/Mal3-143 MTs showing the B-lattice interdimer binding sites occupied by Mal3 (purple) but missing from the A-lattice seam contacts. α-tubulin is shown in dark green and β-tubulin in light green. **b** Close-up portion of the central β-sheet of β-tubulin and an α-helix/sheet region of α-tubulin illustrating the quality of the symmetrized reconstruction

Mal3-143 + Sp_tub EM density shows a good match with our cryo-EM density at the E-site (Fig. 4e, Supplementary Fig. 5c). Together with difference density attributable to the nucleotides at each site (Supplementary Fig. 5d), our structure is thus consistent with GTP hydrolysis having occurred at the Mal3-143 + Sp_tub E-site. This conclusion is supported by conservation of residues around the active site (Supplementary Fig. 2, 3) and with the idea of EB family members acting as tubulin GAPs[20]. A wider view of the Mal3-143 + Sp_tub and EB + Mam_tub structures aligned at the E-site, suggests instead that the compaction within the PF that accompanies GTP hydrolysis in the EB + Mam_tub MTs does not occur in the Mal3-143 + Sp_tub MTs[12]. We attribute this to intrinsic differences in the Sp_tub protein, as highlighted by the relative positions of the mechanochemically important α-tubulin H7 (Fig. 4f, red asterisk). We therefore propose that Sp_tub is intrinsically non-compactable, perhaps due to the tilted conformation of the Sp_tub dimers, thereby altering the coupling between the biochemical state of the Sp_tub dimers and MT lattice structural state. This is in marked contrast to the property of Mam_tub MTs where lattice compaction is coupled to the hydrolysis of GTP.

**Structure of lateral interprotofilament contacts**. On the lumenal side of the MT wall, lateral contacts between neighboring tubulin monomers connect the PFs. α- and β-tubulin are readily distinguished in this view by the characteristic eight-residue insertion in the α-tubulin S9–S10 loop (Fig. 5a, Supplementary Fig. 2, Supplementary Movie 1). In addition, density corresponding to the H1–H2′ loop in α-tubulin is less defined compared to β-tubulin (Fig. 5a, red asterisk). In mammalian α-tubulin, this 19-residue loop contains Lys40, which is subject to acetylation. The structure of this loop has not been visualized in any mammalian MT reconstruction (e.g., ± acetylation[12,23,42,43]), presumably due to intrinsic flexibility. The Sp_tub used in our reconstructions contains no post-translational modifications[28], nor any lysine residue in this region that could be a potential substrate for acetylation. However, our reconstruction uses a wild-type mixture of Sp_α-tubulin1/2 isoforms (87% sequence identity overall; Supplementary Fig. 2): in α-tubulin1, the H1–H2′ loop is a 13-

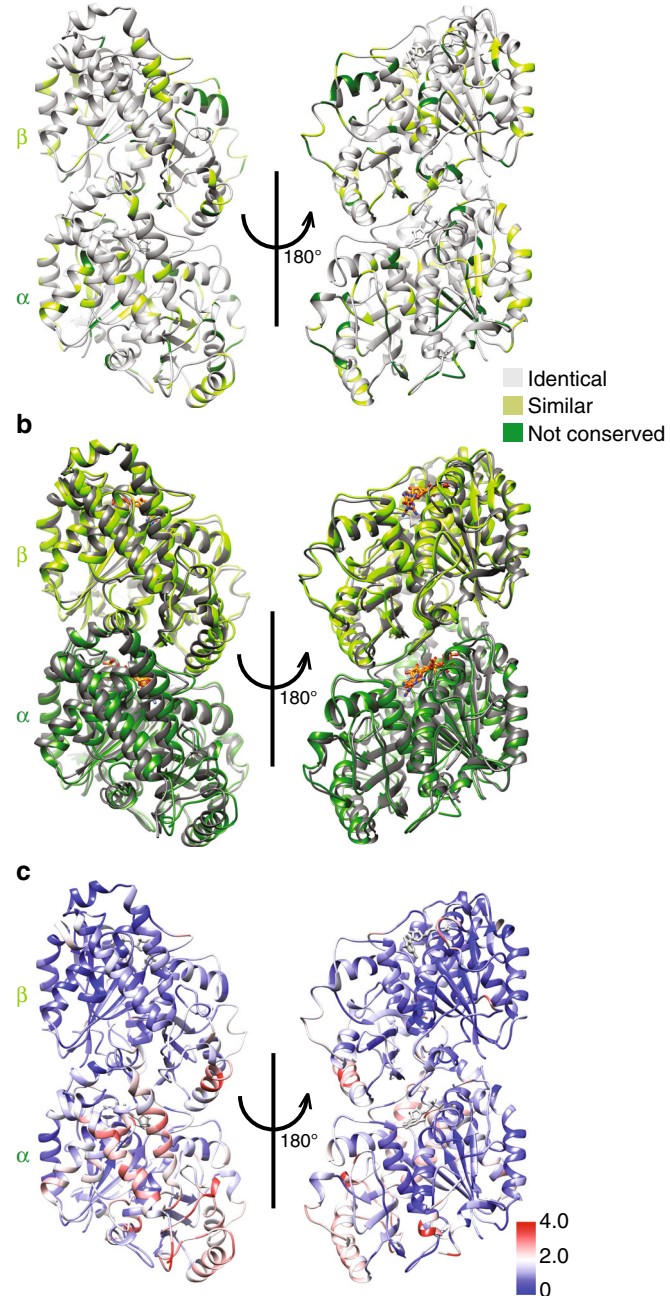

**Fig. 3** Structure of Sp_tub dimer. **a** The Sp_tub dimer structure viewed from the MT outside (left) and inside (right) colored according to sequence conservation with mammalian tubulin. **b** The Sp_tub dimer structure (light green/dark green) overlaid on mammalian tubulin (gray) aligned on the central β-sheet of β-tubulin (PDB 3JAR) viewed from the MT outside (left) and inside (right). **c** The Sp_tub dimer structure colored according to structural differences (RMSD/Å) compared to mammalian tubulin as depicted in viewed from the MT outside (left) and inside (right)

residue loop, while it is nine residues in Sp_α-tubulin2. The mixture of α-tubulin protein is evident from the poorer density and resolution in this region of the reconstruction (Supplementary Fig. 1c) and we therefore did not build this loop in our molecular model. However, this is only a local perturbation and the two isoforms adopt a highly similar conformation throughout the rest of the α-tubulin monomer, allowing its structure to be

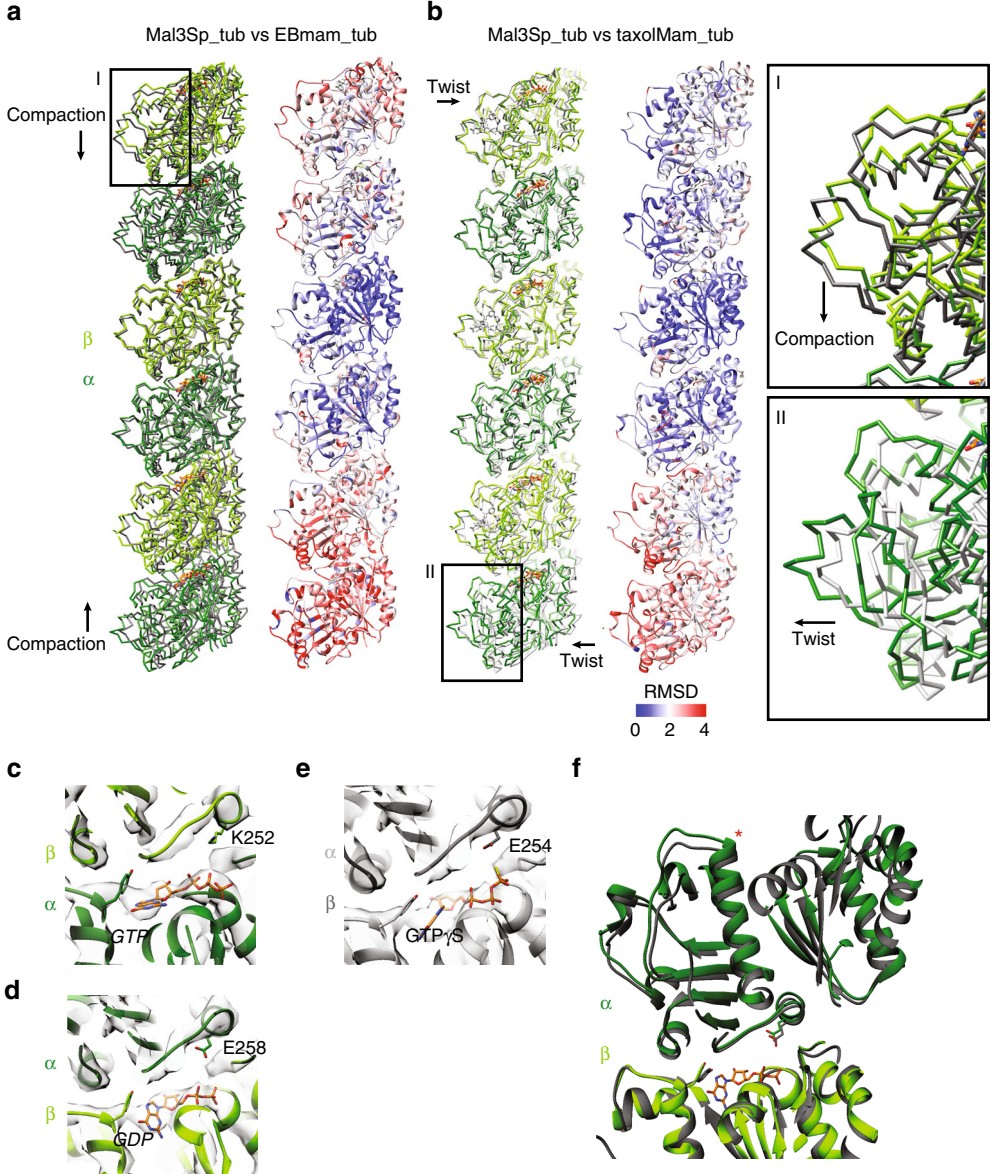

**Fig. 4** Structure of longitudinal Sp_tub PFs reveals an extended tubulin repeat while supporting GTP hydrolysis. **a** Overlay (left) and RMSD (right) of 3 dimers from a Mal3 + Sp_tub PF (green) with the EB3 + Mam_tub (gray) PF aligned on the central β-sheet of β-tubulin in the middle tubulin dimer, showing lack of the longitudinal PF compaction in Mal3 + Sp_tub that is present in EB3 + Mam_tub. **b** Overlay (left) and RMSD (middle) of 3 dimers from a Mal3 + Sp_tub (green) PF with the Taxol + Mam_tub (gray) PF aligned on the central β-sheet of β-tubulin in the middle tubulin dimer, showing a similarly extended PF structure in each. This also shows the intrinsic skew of the Mal3 + Sp_tub PFs that is not exhibited in the Taxol + Mam_tub MTs. Right, box I, zoomed view of Mal3 + Sp_tub vs EB3 + mam_tub overlay, indicating the direction of nucleotide-dependent compaction seen in EB3 + mam_tub but absent in Mal3 + Sp_tub. Box II, zoomed view of Mal3 + Sp_tub vs Taxol + Mam_tub overlay, indicating the direction of lattice twist in Mal3 + Sp_tub relative to Taxol + Mam_tub. **c** Mal3 + Sp_tub α-tubulin N-site with Sp_tub molecular model docked (dark green). **d** Mal3 + Sp_tub β-tubulin E-site with Sp_tub molecular model docked (light green). **e** Mal3 + Sp_tub β-tubulin E-site with EB3 + Mam_tub molecular model docked (gray) [+ GTPγS]. **f** Overlay of the E-site of Mal3 + Sp_tub (green) with the EB3 + Mam_tub (gray) aligned on the E-site, showing lack of the longitudinal dimer compaction in Mal3 + Sp_tub that is present in EB3 + Mam_tub, highlighted by the difference in positions of α-tubulin H7 (red asterisk)

visualized at near-atomic resolution (Supplementary Fig. 1c). For simplicity, numbering of residues for α-tubulin1 is used below.

The loops that form the lateral contacts are the same in both α- and β-tubulin and in mammalian and yeast MTs: loops H1′–S2 and H2–S3 protrude together from one side of each monomer and meet the so-called M-loop (S7–H9) from the adjacent monomer (Fig. 5b,c). The structures of the individual lateral B-lattice contacts in our Sp_tub reconstruction are conformationally very similar to those seen in mammalian MTs[23]; also supported by PISA calculations of both lateral interfaces (see Methods), which themselves are reported to adopt the same conformation

independent of the nucleotide/ligand bound[12]. Thus, the differences in quarternary organization of the PFs in Sp_tub MTs (Fig. 1) compared to mammalian MTs are flexibly accommodated by structurally very similar lateral contacts, as was previously observed for different mammalian MT architectures[44].

Aromatic residues are found in the middle of the M-loop of each subunit and are located in density between the H1′–S2/ H2–S3 loops from the laterally adjacent monomer, as was seen in mammalian MTs[12]. In Sp_α-tub, this is Sp_His287 (mam_-His283, Fig. 5b, Supplementary Fig. 2). In Sp_β-tubulin, Phe281

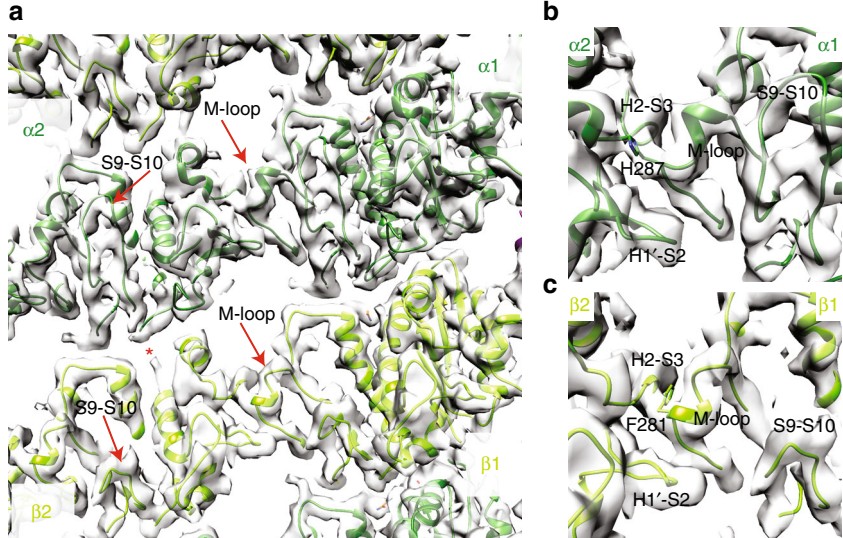

**Fig. 5** Structure of lateral inter-protofilament contacts in *S. pombe* MTs. **a** View of inter-PF lateral contacts viewed from the MT lumen, with key secondary structure features highlighted. The S9–S10 loop is 8 residues longer in α-tubulin allowing ready differentiation between α- and β-tubulin. Density corresponding to the H1–H2′ loop in α-tubulin, which is less well defined compared to β-tubulin, is indicated with a red asterisk. **b** Close up of α/α-tubulin lateral contacts. **c** Close up of β/β-tubulin lateral contacts

forms the key connections with lateral loops from the adjacent monomer (Fig. 5c, Supplementary Fig. 3). The equivalent residue in mammalian tubulin is Tyr281, suggesting that the dynamics of Sp_β-tubulin lateral contacts may be altered by the inability of Phe281 to a hydrogen bond with Sp_Asn85 (mammalian Gln85, Supplementary Fig. 3).

Other *S. pombe*-specific sequence substitutions are located at the lateral contacts, although side chain conformations for these residues are resolution limited in our reconstruction. In α/α-tubulin lateral contacts, the residues within the lateral contacts themselves are almost completely conserved between two α-tubulin isoforms. In the H1′–S2 loop, Gln62 (Mam_Ala58) would be directed toward the adjacent M-loop and in the H2–S3 loop, Pro86 (Mam_Thr82) may constrain the specific conformation of this loop (Supplementary Fig. 2). The lateral contacts formed by the single β-tubulin isoform adopt a very similar conformation as the α-tubulin lateral contacts, with residues from the H1–H1′ loop clearly connected to the H1′–S2 lateral loop (Supplementary Fig. 5). However, none of the *S. pombe*-specific substitutions in these loops make obvious direct contributions to the lateral interface. Similarly, substitutions in the region of the H2–S3 loop are not located directly at the lateral interface (Supplementary Fig. 3). However, of the 14 residues in the β-tubulin M-loop, nine are substituted in *S. pombe* compared to the mammalian sequence, six of which are nonconservative substitutions (Supplementary Fig. 3). Thus, while most substitutions in the loops that form lateral contacts of both monomers appear less likely to directly impact the lateral interface, allosteric effects due to substitutions in neighboring structural elements are possible.

**The binding site for Mal3 on Sp_tub MTs.** The assignment of α- and β-tubulin within the MT lattice allowed us to conclude that the Mal3 CH domain binds at the corner of four tubulin dimers, as has been previously described[20,23] (Fig. 6a, Supplementary Movie 1). The overall disposition of the Mal3 CH domain on Sp_tub MTs is very similar to Mal3[20] and EB3[23] bound to B-lattice mammalian MTs. Thus, for example, helix-α1 of the CH domain lies close to β-tub2, helix-α5 contacts α-tub2, while the CH domain C terminus emerges from the front face of the CH domain directed away from the MT, from where it would connect to the rest of the Mal3 molecule, as previously observed (Fig. 6a). This shows that the Mal3/EB family CH-fold and its MT inter-actions are overall well conserved. However, the N terminus of Mal3 is 15 residues shorter than mammalian EB3 (Supplementary Fig. 6a) and, consequently, the interaction between EB3 N-terminal residues and the α2/β2-tubulin cleft[23], is not present in the Mal3-143 + Sp_tub MT complex.

To further investigate Mal3-Sp_tub interaction, we characterized the footprint of the CH domain on the MT surface using PISA[45]. The contacts made with tubulin by the Mal3 CH domain are located at the corners of a rectangular binding interface (Fig. 6b,c, Supplementary Fig. 6a) and are formed with all four tubulin heterodimers at the Mal3-binding site. The diagonally related contacts with α-tub1 and β-tub2 appear more substantial and, in the case of the β-tub2 contacts, correspond to surfaces of CH domain with more basic residues (Fig. 6c, bottom). Intriguingly, the footprint of Mal3 on Sp_tub MTs is not the same as EB3 on mammalian tubulin MTs (Supplementary Fig. 6b, PDB: 3JAR[23]), as it is also reflected in the difference in charge distribution of EB3 at the MT interface (Fig. 6c bottom and Supplementary Fig. 6b right). Thus, despite conservation of the overall MT-binding site, conservation of the lattice interaction of Mal3 compared to EB is high but not complete.

To investigate this further, we also analyzed the binding interface of Mal3 with GTPγS-Mam_tub MTs (Supplementary Fig. 6c[20]). This shows that the previously characterized region of contact between Mal3 and GTPγS-Mam_tub MTs and Sp_tub MTs is similar (Supplementary Fig. 6c). We also generated a structural model of Sp_tub MT in a compacted-like conformation (Supplementary Fig. 6d) to investigate how the Mal3 footprint would be altered as a result of such a putative conformational changes in the MT lattice. In this model, the Mal3-binding residues identified in our cryo-EM reconstruction are shifted such that their disposition with respect to their binding partners on Mal3 is imperfectly aligned (Supplementary Fig. 6d). This suggests that if compaction across two interdimer interfaces at the Mal3-binding site were to occur in the Sp_tub MT lattice, Mal3-binding would be perturbed, in contrast to the effect of compaction in alpha tubulin close to the heterodimer interface upon EB3 binding to Mam_tub MTs in which EB3 binding is favored.

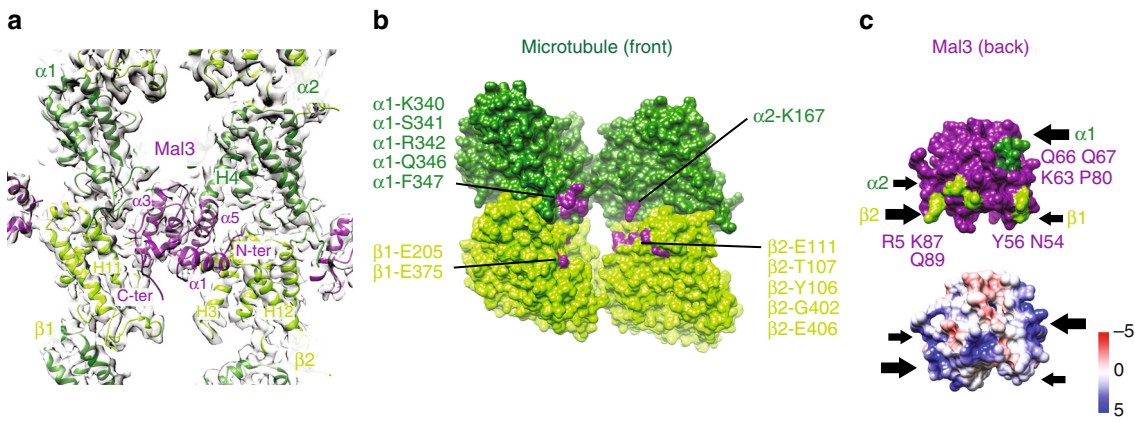

**Fig. 6** Consequences of *S. pombe* MT architecture for Mal3 binding. **a** View from the outside of the Sp_tub MT of the Mal3 binding site at the corner of four tubulin dimers, with contacts made to α-tub1 and α-tub2 (dark green), β-tub1 and β-tub2 (light green). **b** Mal3 footprint on the MT surface with polar and charged tubulin residues < 4 Å distant from Mal3 colored purple and labeled. **c** MT-binding surface of Mal3; top, polar and charged Mal3 residues < 4 Å distant from the MT colored in green and labeled; bottom, charge distribution of the Mal3 MT-binding surface. The black arrows indicate the four contact points of Mal3 with the MT

**Binding of Mal3 to dynamic *S. pombe* and brain MTs**. Having shown that Mal3 recognizes Sp_tub differently to Mam_tub MTs, we sought to compare the in vitro tip-tracking behavior of Mal3 on Sp_tub MTs vs. Mam_tub MTs. To do this, we used two-color TIRF microscopy to visualize the binding of dimeric Mal3 or EB1 carrying a C-terminal e-GFP label to dynamic MTs polymerized from either Sp_tub or Mam_tub (Fig. 7). Under our standard assay conditions (PEM100 + 1 mM GTP), Mal3-GFP binds uniformly to the tip and lattice of Sp_tub MTs (0.5 nM–50 nM; Fig. 7b). In this same buffer, EB1-GFP also binds uniformly to Sp_tub MTs, albeit with substantially higher concentrations of EB1-GFP being required to drive binding (100–500 nM; Fig. 7c). By contrast, on Mam_tub MTs, 50 nM Mal3 preferentially binds the growing MT plus ends, although clearly, there is still binding to the lattice (Fig. 7d). Increasing the KCl concentration depopulates the lattice more than it does the tips of Mam_tub MTs (Fig. 7d), further emphasizing that Mal3 binds differently to the tips of Mam_tub MTs compared to the lattice. With Sp_tub MTs also, adding KCl to the buffer depopulates the lattice of 50 nM Mal3-GFP but has much less effect on the tip-bound population, again producing the classic 'tip-tracking comet' appearance (Fig. 7e).

These data show first that both Mal3 and EB1 bind the growing ends of dynamic MTs more tightly than they bind the lattice, but second that the difference in occupancy between the tip and lattice is very much less obvious with dynamic Sp_tub MTs than with dynamic Mam_tub MTs. By carefully titrating the buffer conditions, we were able to depopulate Mal3 from the MT lattice while leaving it in place at the tips of Sp_tub MTs (Fig. 7e). Under "more usual" conditions, this difference was not apparent (Fig. 7b), because Mal3 occupancies at the tips and lattice were approximately equal. These data emphasize that when matched to its canonical MTs, Mal3 is less able to differentiate between the (presumably) GTP-rich tip and the (presumably) GDP-rich lattice conformations of MTs. Our cryo-EM data provide a structural explanation for this behavior by showing that in *S. pombe* MTs, the transition from GTP state to GDP state produces much less lattice compaction than that which is seen for mammalian MTs.

## Discussion

Dynamic instability is a fundamental property of the MT cytoskeleton. The structural mechanisms linking filament dynamics and the tubulin GTPase are being gradually uncovered, with significant recent progress in cryo-EM allowing mechanistic dissection of the conformational responses of mammalian tubulin to different nucleotide states, the presence of stabilizing drugs, and the binding of regulatory proteins[23,43,46]. However, whether tight coupling between the bound ligand and the conformation of the MT polymer is the rule or the exception is not yet known. Here, by revealing that 1) unlike Mam_tub MTs, Sp_tub MTs do not significantly change their lattice spacing in response to a nucleotide and 2) that the cognate EB decreases the PF skew of Sp_tub MTs while increasing the skew of Mam_tub MTs, we have uncovered an underlying structural plasticity in MT assembly, showing that MT lattice conformation is sensitive to a variety of effectors, and differently so for different tubulins.

Our cryo-ET data enabled direct observation of in vitro-polymerized *S. pombe* MTs without imposing assumptions about their architectures or performing any computational averaging. Specifically, these data allowed visualization of the intrinsic PF skew within Sp tub MTs, even in 13-PF MTs, and particularly in the presence of GMPCPP (Fig. 1, Table 1). PF skew is present in Mam_tub MTs but to a much lesser extent, and this property of Sp_tub MTs is likely to influence their interaction with, and regulation by, binding partners.

Strikingly, our visualization of Sp_tub MTs using cryo-ET provided no evidence that Mal3 can drive formation of MTs with extra seams or entirely A-lattice MTs, as previously described[19]. An earlier study using low-resolution metal shadowing[21] also shows Mal3 binding only to the A-lattice seams of mammalian brain MTs. However, more recent work[20,23] and our present study show completely the opposite—that Mal3/EBs avoid A-lattice seams. While all relevant studies agree that coassembly with Mal3 promotes nucleation and growth of 13-PF MTs, with additional evidence for stabilization of GDP MTs by Mal3[29], the discrepancy concerning seam binding by Mal3 and the architecture of the resulting MTs is stark and we have not yet discovered its origin. One possible explanation for our not finding MTs with extra seams in the present work is that MTs produced by coassembly with Mal3 are unstable[22]. Further, all higher-resolution structural studies of Mal3/EBs to date have used monomeric EB constructs (this work and[20,23]), while the effect of the native, dimeric state of Mal3/EBs on their MT lattice recognition—as used in the above-described metal-shadowing experiment[21]—has not yet been studied by high-resolution EM. Investigating this point will require further work and we are actively engaged in this. What is clear at present is that under the

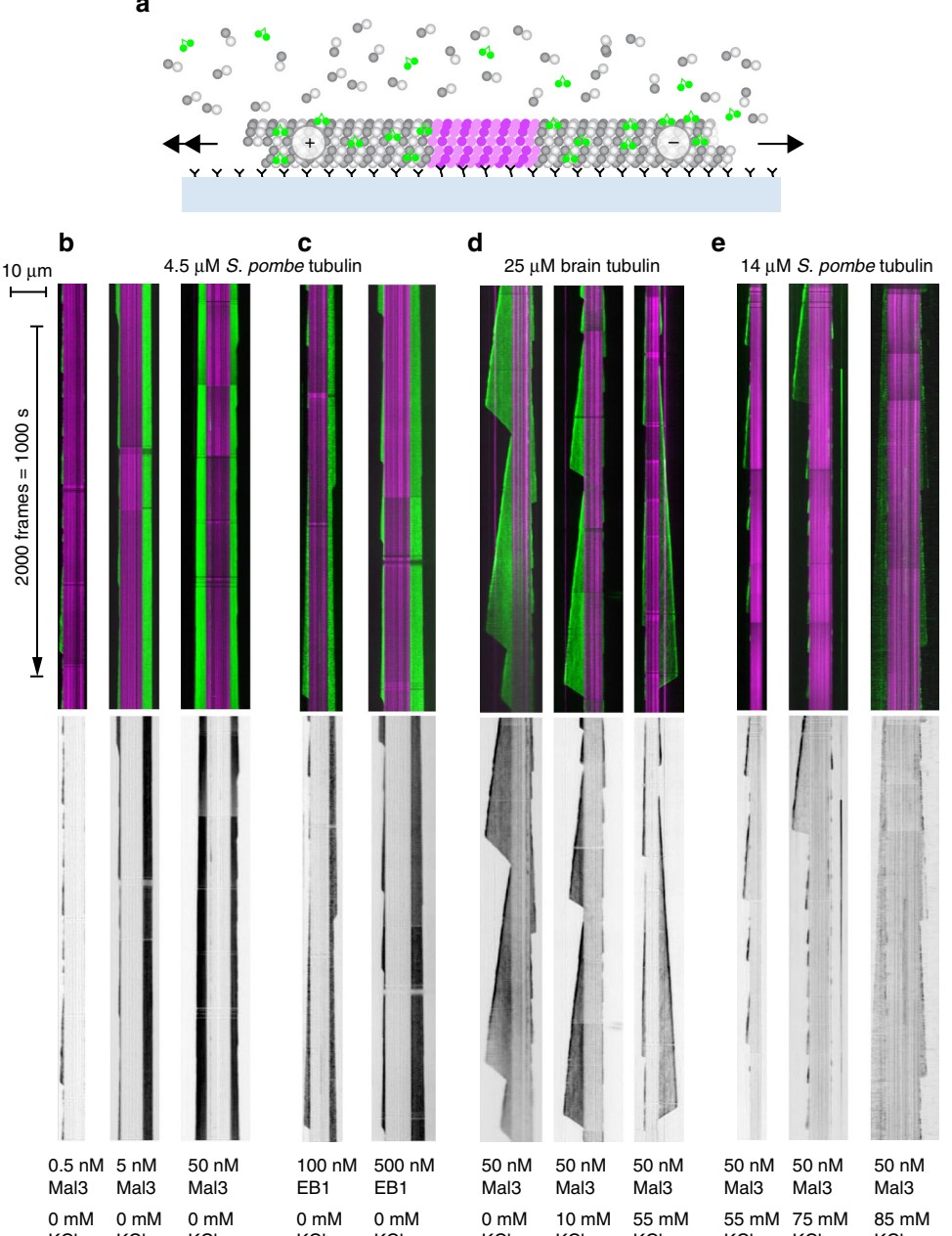

**Fig. 7** TIRF microscopy of dynamic MTs + Mal3/EB. **a** Schematic of experimental setup. Dynamic MTs are grown from stabilized seeds (magenta) linked to the glass surface by anti-fluorophore antibodies. Binding of EB1-GFP or Mal3-GFP (green) is visualized using TIRF microscopy. **b–e** Kymographs showing the pattern of Mal3-GFP binding or EB1-GFP binding (green) to dynamic, unlabeled Sp_tub MTs or Mam_tub MTs. Depending on the buffer conditions and protein concentrations, differential binding to tip and lattice (tip-tracking) is or is not apparent. **b** Mal3-GFP binding (green) to dynamic, unlabeled Sp_tub MTs grown at 4.5 μM Sp_tub concentration from antibody-immobilized, Alexa-488-labeled GMPCPP-Mam_tub seeds (magenta). Buffer is PEM100 plus 1 mM GTP. Greyscale kymograph shows the Mal3-GFP channel only. Under these conditions, Mal3-GFP binds uniformly to the lattice, without obviously favouring the MT tips. **c** Kymographs showing the pattern of EB1-GFP binding (green) to dynamic, unlabeled Sp_tub MTs grown at 4.5 μM tubulin concentration from antibody-immobilized, Alexa488-labeled GMPCPP-Mam_tub seeds (magenta). Greyscale kymograph shows the EB1-GFP channel only. Buffer is PEM100 (see Methods section) plus 1 mM GTP. Under these conditions also, EB1-GFP binds uniformly to the tips and the lattice. **d** Kymographs showing the pattern of Mal3-GFP binding (green) to dynamic, unlabeled Mam_tub (pig brain) MTs grown at 25 μM tubulin concentration from antibody-immobilized GMPCPP-Mam_tub seeds (magenta). Greyscale kymograph shows the Mal3-GFP channel only. Buffer is PEM100 plus 1 mM GTP plus additional KCl as indicated. Under these conditions, Mal3-GFP enriches at growing plus ends. Raising the KCL concentration depopulates the lattice more than the tips. **e** Kymographs showing the pattern of Mal3-GFP binding (green) to dynamic, unlabeled Sp_tub MTs grown at 14 μM tubulin concentration from antibody-immobilized GMPCPP-Mam_tub seeds (magenta). Greyscale kymograph shows the Mal3-GFP channel only. Buffer is PEM100 plus 1 mM GTP plus additional KCl as indicated. Mal3-GFP enriches at growing plus ends. Under these conditions, plus end assembly is frequently interrupted by catastrophe

conditions used here, coassembly with 25 µM monomeric Mal3 converges the architecture of *S. pombe* MTs toward a canonical B-lattice-based 13_3 symmetry with a shallow PF skew and a single A-lattice seam.

Our cryo-ET data also provided a route to unbiased validation of structural parameters prior to high-resolution structure determination. Using this information and a single-particle averaging approach, we obtained a 4.6-Å structure of Sp_tub MTs formed in the presence of GTP and Mal3. This is the first near-atomic resolution structure of Sp_tub MTs. We observed similar lateral interactions in our structure compared to those in mammalian brain tubulin, even though the tubulin sequences are not identical in these loops. The near-atomic resolution reconstructions of EB3 + Mam_tub MTs with different nucleotides reveal that the EB-driven conformation of MTs is compacted, with a small PF skew[23]. This state is proposed to be post hydrolysis, but with the $_{Pi}$ still in the E-site. Thus, the current concept for the structural changes within the lattice that accompany the GTPase of mammalian tubulin is a three-step model: 1) the "GTP-bound" state, which shows extended MTs without a PF skew; 2) the "post-hydrolysis" (GDP.Pi) state, which describes compacted MTs with a PF skew; and 3) the "GDP" state, which is formed by compacted MTs without a PF skew. Our observations regarding the longitudinal lattice spacing show differences to this model in the behavior of Sp_tub MTs: independent of the assembly conditions used, we only observed Sp_tub MTs in an extended conformation. In addition, we show that the intrinsic skew in Sp_tub MTs is similar to that seen in EB−MTs + GTPγS. Thus, although the PF skew that we observed for Sp_tub is nucleotide dependent, its extended longitudinal spacing appears to be independent of the nucleotide bound at the E-site. This conformation——with extended longitudinal spacing and skewed PFs—— has not previously been observed as part of the mammalian MT GTPase cycle.

Importantly, our structure enabled us to analyze the binding interface between Sp_tub MTs and its in vivo-relevant ligand Mal3. Mal3 binds between four heterodimers of the Sp_tub MT B-lattice, as previously observed in lower-resolution studies of Mal3 with Mam_tub MTs[20] and the recent high-resolution study of EB3 with Mam_tub MTs[23]. Unsurprisingly, given this shared binding site, a number of MT-binding residues are common between Mal3 and EB3 and are therefore likely to be conserved in all eukaryotes (Supplementary Fig. 6A). Since Mal3 stabilizes 13-PF Sp_tub MTs with a shallow skew, and EB3 tends to converge brain MTs to a 13-PF arrangement with a shallow skew—but in this case by increasing the intrinsic skew—this may partly explain this shared binding site. However, when comparing the Mal3-Sp_tub MT interaction in our structure with previous structures of MT-bound EB/Mal3, the precise MT-binding interface of each protein is distinct. While previous work has shown that EBs drive substantial lattice compaction in mammalian brain MTs, Mal3 induces only very slight lattice compaction in *S. pombe* MTs. Together, these data indicate that while Mal3 binding stabilizes the GDP lattice of both brain MTs and *S. pombe* MTs, the corresponding conformational change only produces substantial lattice compaction in mammalian MTs and is mediated by a different sensitivity to the biochemical state (GTPγS) of the mammalian tubulin.

A recent investigation of the structure of *S. cerevisiae* tubulin (Sc_tub) and its interaction with the *S. cerevisiae* EB, Bim1, provides a further insightful perspective on structural mechanisms of EBs[47]. These data, together with an earlier study[48], indicate that, like Sp_tub, the polymerization properties of Sc_tub are different from Mam_tub and that Sc_tub also tends to form a lower number of PF MTs with some PF skew. However, Sc_tub appears less susceptible to modulation of PF skew by tubulin

nucleotide than we observed in Sp_tub (Fig. 1c). In the absence of Bim1, Sc_tub also does not undergo the nucleotide-dependent lattice compaction seen in Mam_tub. However, Bim1 binding does cause compaction in Sc_tub MTs. This difference could be related to Bim1's binding pattern on the Sc_tub MT lattice: while the CH domain of each protein has the same orientation with respect to the MT surface between PFs, whereas Mal3 (and mammalian EB3) binds to every tubulin dimer, Bim1 binds in Sc_tub MTs to every tubulin monomer. Alignment of the CH domain sequences of Bim1 and Mal3 (Supplementary Fig. 7a) highlights differences between the proteins that could contribute to this difference[47]. Differences in the specifics of the samples used in each study—construct sizes for Mal3 (1-143) vs. Bim1 (1-210 + EGFP), use of tagged (Sc_tub) or untagged (Sp_tub) tubulin, and use of GTPγS in the Sc_tub-Bim1 complex formation vs. GTP in Sp_tub-Mal3 complex—could also contribute to the differences in binding pattern, and might contribute to the difference in resolution in these structures (Sp_tub-Mal3 = 4.6 Å; Sc_tub-Bim1 ~10 Å). However, this comparison also raises intriguing questions about the conservation or lack of it in MT−EB interactions. Future work will aim to dissect the molecular basis of these differences.

Previous studies of Mal3 in vitro have shown that it selectively accumulates at growing ends of dynamic Mam_tub MTs while binding more weakly along the entire length of MTs[15,16]. We see Mal3 binding all over the MT lattice in cryo-EM experiments and in TIRF. In low-salt conditions used for cryo-EM experiments, EB1 was also observed to bind along the MT lattice ([24] and Fig. 7a). However, here, we show that even at elevated salt levels, Mal3 binds to the Sp_tub MT lattice (Fig. 7d). Therefore, in dynamic MTs, Mal3 may recognize the underlying skew of Sp_tub MTs over and above any preference for nucleotide-dependent structural states of tubulin (if they exist) in Sp_tub MTs. Our data show that Mal3 has affinity for the Sp_tub MT lattice as well as its ends, inhibits the disassembly of Sp_tub MTs (Fig. 7b,c[29]), and therefore may have a broader function of stabilizing the lattice in *S. pombe* beyond tracking the MT tip, an idea supported by data in vivo[35].

Nevertheless, it is clear that Mal3 can recognize the differences between multiple regions on MTs. Besides being able to distinguish A-lattice PF interfaces (seams) from B-lattice PF interfaces, Mal3 can also distinguish the tip from the lattice in dynamic Sp_tub MTs, albeit with less sensitivity than mammalian proteins. Since our data indicate that there may be no mechanochemical response within the Sp_tub MT lattice to the tubulin GTPase cycle, the difference between the tip and the lattice may be due to the existence of non-lattice-like polymeric conformation of the MT tip not accessible in our high-resolution structural experiments and as also observed previously for mammalian EBs[24]. In addition, Mal3 tip tracking in vivo is well documented[34,49,50]. Given the structural differences of Mal3 response in vitro, it is likely that additional binding partners such as Dis1[51] and Alp7[30] help tip-tracking specificity of Mal3 in vivo and that the collaboration between Mal3 and other tip-binding proteins in yeast is different in comparison to the mammalian system. Mal3 is also subject to cell cycle-dependent phosphorylation that may alter its sensitivity to particular binding sites within the MT cytoskeleton[34,50]. Taken together, these data show that MT dynamics and MT binding by EB are different in different species, and findings from one system cannot necessarily be extrapolated to other systems. It will be desirable to widen the focus of structural and functional studies on tubulin to examine a larger variety of species in the future. Organisms and cell types with PF numbers other than the canonical 13 PFs will be of particular interest in this context[52].

Our data reveal that MTs built from *S. pombe* tubulin have a different architecture to those built from mammalian brain tubulin, and that coassembly with the respective EB family binding partners serves to focus both systems toward a common 13_3 B-lattice structure with slightly tilted PFs and a single A-lattice seam[23]. To do this, EBs need to exert distinctly different effects on *S. pombe* vs. brain MTs—*S. pombe* MTs need to have their PF twist decreased but require little or no additional lattice compaction, whereas brain MTs need to have their PF twist increased slightly, and their lattice compacted. By showing that Mal3 has distinct structural effects on *S. pombe* vs. brain MTs, both of which are dynamically unstable, our data support a structural plasticity model[53] in which MT lattice conformation is sensitive to a variety of effectors and differently so for different tubulins. In tight-coupled models of dynamic instability, tubulin conformation is serially switched by the turnover of GTP in the tubulin active site. By contrast, in structural plasticity models, the dynamics of the polymer are still coupled to the kinetics of nucleotide turnover, but the coupling is looser, allowing each nucleotide state of tubulin to adopt a range of different conformations while still preserving the underlying thermodynamic drive for dynamic instability provided by nucleotide turnover. Overall, our data support a more nuanced "subtle allostery"[54] picture for the relationship of microtubule lattice conformation to the chemical state.

## Methods

**Tubulin preparation**. Untagged wild-type *S. pombe* tubulin (dual isoform) and single-isoform (in which the nonessential α-tubulin2 gene has been replaced by a second copy of the α-tubulin1 gene) *S. pombe* tubulin proteins were expressed and biochemically purified, as previously described in detail[28]. In brief, 80-l cultures of *S. pombe* were grown in a fermenter, collected by centrifugation, resuspended, and broken open using a bead mill. Tubulin in the soluble cell fraction was purified using ion exchange chromatography, ammonium sulfate precipitation, a cycle of polymerization/depolymerization, and size-exclusion chromatography. Purification procedures were the same for the single- and double-isoform tubulins, with only the starting cells differing.

**Sample preparation for electron microscopy**. Monomeric Mal3-143 was expressed in *E. coli* and purified as previously described[19]. *S. pombe* MTs were assembled from tag-free, dual-isoform-purified endogenous tubulin in PEM buffer (100 mM PIPES-KOH, 1 mM MgSO$_4$, and 2 mM EGTA, adjusted to pH 6.9 with KOH)[28] and mixed 1:1 with MES buffer (100 mM MES, pH 6.5, 1 mM MgCl$_2$, 1 mM EGTA, and 1 mM DTT) at 32 °C for all conditions except for the dynamic GTP MTs. For dynamic GTP MTs, 60 μM tubulin was polymerized in the presence of 5 mM GTP in PEM for 1 h. For GMPCPP MTs, 20 or 30 μM tubulin was polymerized in the presence of 1 mM GMPCPP (Jena Bioscience). For GTP-Mal3 MTs, 20 or 30 μM tubulin was polymerized in the presence of 5 mM GTP and 25 μM Mal3. For Mal3 + GMPCPP-Sp_tub MTs, 20 or 30 μM tubulin was polymerized in the presence of 1 mM GMPCPP and 25 μM Mal3 for 1.5 h.

**Cryo-EM grid preparation and data collection**. The Cut7 kinesin motor domain (KMD, residues 1-432) was expressed and purified as previously described[55], and buffer exchanged into 25 mM K-Pipes, pH 6.8, 30 mM NaCl, 5 mM MgCl$_2$, 0.5 mM EGTA, 1 mM 2-mercaptoethanol, and 5 mM AMPPNP prior to cryo-grid preparation. A concentration of 13 μM *S. pombe* MT populations polymerized under the four different conditions described above were incubated with either 60 μM Mal3, or with 24 μM KMD, and 4 μl of the mixtures were applied to glow-discharged Quantifoil R 2/2 holey carbon grids which were blotted and plunge frozen into liquid ethane using a Vitrobot IV (FEI) operating at room temperature and 100% humidity. Samples used for cryo-electron tomography also contained 10 nm Protein-A-coated gold fiducial markers (Electron Microscopy Sciences).

Single-axis cryo-EM tomograms were recorded using a Polara microscope operating at 300 kV on a Gatan K2 direct electron camera in counting mode, with a quantum energy filter, using a pixel size of 3.5 Å/px. Tilt series were acquired from −60 to +60 degree using a total dose of 40–60 e⁻/Å$^2$ and a 3-degree step using 2-s exposure and a frame rate of five frames per second per tilt, and a −5-μm defocus. Tilt series were aligned, CTF corrected, reconstructed, and filtered with an anisotropic filter using a $k = 15$–40 and 10–30 iterations in IMOD[55].

Cryo-EM data for single-particle reconstruction was collected at 300 kV on a Polara and a Gatan K2 direct electron camera in counting mode, with a quantum energy filter, recording in total 994 movies with a total dose of 30 e⁻/Å$^2$ fractioned

into 50 frames at a pixel size of 1.39 Å/px. Initial frame alignment was done with IMOD[56]. A second local alignment step was performed with Scipion[57] using the optical flow method. In the final reconstruction, only frames 2–21 were included resulting in a total dose of 12 e⁻/Å$^2$.

**MT architecture analysis**. MT subvolumes were extracted from 3D tomograms and their transverse sections were extracted computationally and subjected to multivariate statistical analysis in IMAGIC using the 'rotate_randomly' command to determine the underlying symmetry and thus the PF architecture of each MT[36]. The PF number of each MT was directly determined by inspection of the resulting eigen images and the corresponding class averages. The 3D MT tomograms were summed along their *Z*-axis and the resulting moiré pattern repeat was measured manually in Fiji[58]. To visualize the organization of the underlying MT lattice, 20–30 longitudinal slices of the upper or lower part of the MT tomograms were summed and inspected for the pattern of KMD decoration on the MT lattice.

**Structure determination of Mal3-bound *S. pombe* MTs**. A total of 27129 MT segments were selected in 908-Å$^2$ boxes in Boxer[59] using the helix option and choosing an overlap that left three tubulin dimers (240 Å) unique in each box. Of the 2466 MTs that were initially boxed from 994 motion-corrected movies, 1070 MTs with 13_3 architecture were selected. The final 3D reconstruction contained 12763 segments, and was calculated using a semiautomated single-particles approach for pseudohelical assemblies in SPIDER and FREALIGN[41]. To avoid model bias, the reference was filtered in each refinement iteration (15 Å in the first and second iteration, to 12 Å in the third and fourth iteration and to 10 Å in the final iteration). The final reconstruction was automatically B-factor sharpened in RELION with an automated calculated B-factor of −237[60]. The final overall resolution of the masked reconstruction was 4.6 Å (0.143 FSC). Local resolution was calculated with Blocres[61].

**Model building**. Homology models for the *S. pombe* αβ-tubulin dimer and Mal3-143 were calculated with MODELLER[62] using previous structures as a reference (PDB: 3J6Gj and 4ABO). The models were fitted into the density as rigid bodies and then adjusted manually using Coot[63], followed by real space refinement in Phenix[64] for which the refinement resolution was set to 4.8 Å. Structures were visualized and measurements of distances, electrostatic potential, and sequence homology were calculated using Chimera[65]. Interactions at the interfaces in the refined model were calculated using PISA[45].

**TIRF microscopy of EB binding to dynamic MTs in vitro**. For *S. pombe* MT experiments, unlabeled single-isoform *S. pombe* tubulin was used. Short Alexa-488-labeled GMPCPP-stabilized pig brain MT seeds were immobilized on coverslips using anti-Alexa-488 antibodies (Invitrogen) and MT assembly initiated by adding unlabeled single-isoform *S. pombe* tubulin in PEM buffer (100 mM PIPES-KOH, 1 mM MgSO$_4$, and 2 mM EGTA, adjusted to pH 6.9 with KOH) with 1 mM GTP and varying KCl concentrations. The growing MTs were imaged using dark-field microscopy, as described previously[22,29]. The tubulin concentration was titrated so that growth and shrinkage were observed from both ends of the MT without spontaneous nucleation in the field of view. After a baseline of 4.5 μM, *S. pombe* tubulin was established, full-length Mal3-GFP (with its histidine tag (His) previously removed by cysteine protease from tobacco etch virus (TEV) cleavage) was added at constant tubulin concentration, and imaged by TIRF microscopy to determine its ability to bind and to tip track *S. pombe* MTs at different ionic strengths. Mal3-GFP was titrated at concentrations ranging from 0.5 to 300 nM (dimer concentration). EB1 binding was tested in the same conditions as Mal3-GFP binding but using 100 and 500 mM EB1-GFP. Experiments on pig brain microtubules used 25 μM tubulin and a constant Mal3-GFP concentration of 50 nM. The dependence of Mal3-GFP binding and tip tracking on ionic strength was tested at KCl concentrations of 55, 75, and 85 mM, again at 50 nM Mal3. For these experiments, the concentration of *S. pombe* tubulin was raised to 14 μM, in order to ensure that an assembly occurred. Two-channel TIRF images were recorded at 2 fps using an Olympus TIRF microscope and capture software, imported into FIJI using the BioFormats converter as two stacks that were then merged, drift corrected using the Manual Drift Correction plug-in in Imagej/FIJI[58], and converted to two-channel kymographs using the Kymograph Builder plug-in.

**Data availability**. The cryo-EM reconstruction that supports the findings of this study has been deposited in the Electron Microscopy Data Bank with accession code 3522. The docked coordinates reported in this paper have been deposited in the Protein Data Bank, www.pdb.org with accession code 5MJS. The data that support the findings of this study are available from the corresponding author upon request.

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

## Acknowledgements

O.v.L. and C.A.M. were supported by Biotechnology and Biological Sciences Research Council (BB/L00190X/1), and thank Dr Dan Clare for technical help with cryo-EM, Dr Claire Naylor for help with model building, and the wider Birkbeck EM community at Birkbeck for helpful discussions. R.C. thanks Dr Anne Straube for the gift of EB1-GFP. N.A.V. was supported by a Warwick Systems Biology DTC studentship, grant number 1090393. R.C. is supported by a Wellcome Trust Senior Investigator Award (grant number 103895/Z/14/Z).

## Author contributions

O.v.L., R.C., and C.A.M. designed the research. D.R.D. and M.K. supplied reagents. O.v.L. and N.A.V. performed the research. O.v.L., R.C., and C.A.M. analyzed the data and wrote the manuscript, with contributions from all authors.

## Additional information

**Competing interests:** The authors declare no competing financial interests.

