## [Peer Review File · Nature Communications]

Reviewers' comments:

Reviewer #1 (Remarks to the Author):

Loeffelholz et al.: Nucleotide- and Mal3-dependent changes in *S. pombe* MTs: a structural plasticity view of MT dynamics

The authors present a very detailed, structural cryo-electron microscopy (cryo-EM) 3D analysis of *S. pombe* tubulin and microtubules decorated with the *S. pombe* Mal3p end-binding protein, a member of the BB1 family. This is the first study of this kind to combine tubulin and Mal3p from their genuine organism (*S. pombe*). This is a high-resolution study that provides crucial new insights into the function of Mal3p and its interaction with tubulin, and especially near-atomic detail of the binding surface. In a way, this is a continuation of the lab's work on that subject (see: Maurer et al., 2012) that will add additional views and interpretation regarding the mode of action in microtubule end-binding proteins in general.

This is a good piece of work and deserves publication in Nature Communications. The results are very interesting and relevant to a broad scientific audience. Hence, I would recommend publication after some point outlined below have been addressed by the authors.

If I'm not mistaken, the Mal3p used here is a C-terminal truncation (as in the Maurer-2012 paper), while others (e.g. Sandblad et al., 2007) worked with a full-length construct. It might be appropriate to acknowledge the, at least SLIGHT possibility that A-lattice recognition, and the results obtained here (and in Maurer et al. 2012) maybe different because of the nature of constructs used. That could be an important issue for function of the native protein, and in vivo and should not be overlooked.

The figure legend of figure 1 is confusing, and probably incomplete:

Figures 1A and 1B look suspiciously like a decoration of microtubules with kinesin motor domains ... all of them. How comes? The binding geometry is very different from the Maurer-2012 paper, and from figure 5, but also, the observed mass of the decorating particles look much larger (i.e. like kinesins) than what one would expect from Mal3p. Mal3p has been found to bridge laterally between adjacent protofilaments, hence their sensitivity for the lattice. The densities shown here do not bridge, but stand out exactly as kinesins do. Since the panels are arranged in columns, how are A and B images related?

Figure 1A GMPCPP shows two examples, the left microtubule is a 12-pf or 14-pf tube due to the left-right symmetry (has to be an even-pf number), the right one is probably a 15-pf tube, or something else with an odd number of protofilaments. Is this described anywhere, and what are the conclusions? The figure reads as if the authors claim these to be exclusively 13-pf tubes.

Figure 2: The result is stunning, but the presentation is not ... For such complex structures the authors should either provide stereo-pairs (works also on screens), or a movie. Figure 2A is too small. Same criticism also applies to figure 3A, B. The differences between sp and mam-tubulin are important, but relatively small and should be better emphasized for the not so frequent viewer of such structures, especially in a journal with a broad audience.

Figure 6B-E: With the exception of E, I'm not sure why the authors see a particular preference for tips ... Looks to me as if the green intensity is everywhere, except at the seeds. Are these three different experiments each? How is the frequent bleaching and sudden recovery of rhodamine explained?

The general discussion about A and B lattices is a little extensive ... I believe, these facts are all well known by now. Could be streamlined

I might be wrong, but I believe the convention in the yeast field is to write proteins like Mal3 as Mal3p ...

Reviewer #2 (Remarks to the Author):

The paper from the Moores lab showed a structure of Mal3-bound *S. pombe* microtubule with at 4.6 Å by cryo-EM with cryo-ET data showing the predominant B-lattice. By performing structural analysis using cryo-electron microscopy (cryo-EM) and in vitro TIRF microscopy assay. The authors showed mainly three things: 1. Mal3 binds to B-lattice of microtubule; 2. Yeast microtubule tubulin lattice does not undergo compaction upon GTP hydrolysis unlike mammalian microtubule; 3. Difference of binding patterns of yeast Mal3 to yeast microtubule from EB1 to mammalian microtubule.

The resolution value authors claim (4.6 Å) seems reasonable to me. The evaluation of the microtubule structure (compaction and skew) were performed properly.

For the first point (EB protein's binding to B-lattice), authors claim that there has been a discrepancy in the field that EB proteins bind to A-lattice or B-lattice microtubule. However, the

previous reports suggested that EB proteins bind to the A-lattice are either based on low-resolution metal shadowing (Sandblad et al., 2006), structural analysis without using 3D reconstitution (Katsuki et al., 2014) or low resolution reconstructed structure (~20 Å) (des Georges et al., 2008). Other higher resolution cryo-EM structures all agree that EB proteins bind to B-lattice microtubule except the seam (Maurer et al., 2012; Zhang et al., 2015). In addition, a paper published very recently in JCB (Howes et al., 2017), which is not cited in this manuscript, also showed that EB protein binds to B-lattice microtubule. Thus, the result showing Mal3 binds to B-lattice microtubule is not so surprising.

In addition, Howes et al. also showed that yeast tubulin lattice does not show compaction as apparent as mammalian tubulin lattice, which compromises significance of the second point of this manuscript. They also discuss about the skew of yeast microtubule lattice. The only difference is that Howes et al. used EB protein Bim1 and tubulin derived from budding yeast instead of fission yeast as in this manuscript.

For the third point, authors used PISA and homology modeling to claim that Mal3 has evolved such that it is favorable for binding to yeast extended microtubule lattice. Rather, authors should include the result of binding pattern of *S. cerevisiae* EB protein Bim1 to microtubule reported in Howes et al. (2017) and compare with their structure to draw more accurate comparison.

For the part of TIRFM assay, it is really hard to make sense the results since tubulin concentrations are different in each experiment (4.5 μM for *S. pombe* tubulin in Fig.6b and c, 25 μM for brain tubulin in Fig.6d and 14 μM for *S. pombe* tubulin in Fig.6e) and KCl concentrations are not consecutive (0-55 mM KCl for Mal3 vs brain tubulin and 55-85 mM KCl for Mal3 vs *S. pombe* tubulin).

The authors' most important claim here is that Mal3 is less able to differentiate between GTP-rich tip and GDP-rich lattice of yeast microtubule since both have similar structures. However, to me, the results appear completely opposite. In Fig.6e, Mal3 is clearly localized at the tip at 55 mM and 75 mM KCl conditions. The difference in Mal3 intensities of tip and lattice appears more prominent compared with the brain tubulin result in Fig.6d which can undergo compaction. If authors are to convince the readers, authors have to; 1, show clearer kymograph (the first kymograph in Fig.6b and e are hard to see since the growing microtubule are too short) and 2, present quantitative data comparing intensities of GFP at the tip and the lattice.

Also, the writing of the current manuscript is not suitable for a general audience. The author has very MT specific introduction, which is hard to read. Therefore, I feel like this paper is either not suitable for publication in Nature Communications or it needs a major revision to differentiate its finding by comparing their structure with structures reported in Howes et al., 2017.

Minor points:

In the Introduction, authors are citing ref 20, 23 and 24 as examples of “higher resolution cryo-EM studies show EB binding to the B-lattice”, but I would not consider citation 24 (Guesdon et al., 2016) as high-resolution structure showing EB binding pattern.

Authors claimed that docking of EB3+Mam_tub E-site structure is a good match with their Mal3-143+Sp_tubu lattice density map, but it is difficult to see from the figure. Different views of Fig. 3c, d and e as Supplementary Figures would be necessary. Also, In Fig.3d, there is a green letter “ α ” on top of GDP molecule. I think this is a mistake happened when authors were preparing the figure.

In the cryo-EM part in the RESULTS, authors are saying that there are two isoforms of α -tubulin as “reconstruction uses a wild-type mixture of Sp_ α -tubulin1/2 isoforms”. However, in the section of TIRF microscopy of Mal3-GFP and Eb1-GFP binding to dynamic MTs in vitro of METHODS, authors are saying that “unlabeled single isoform S. pombe tubulin” were used. I could not find mentions about purifying single isoform S. pombe tubulin in the METHODS, but what is the difference?

Other minor issues:

For Fig.3f legend, I think it’s common to write like “red asterisk” instead of writing like “red *”. Same with the legend of Fig.4 and legend of Supplementary Fig.5. Also for third paragraph (Fig. 3f, red*) and fourth paragraph (Fig4a, *) in p.7.

Citation format is different only at the last part of the second paragraph in p.5.

“and are consistent with the PFs in Sp_tub MTs being markedly more skewed within the lattice (19).”

There are places without space between numbers and units throughout the manuscript.

In the section of Cryo-EM grid preparation and data collection in the METHODS, it is “holey carbon grids”, not “holy carbon grids”.

Reviewer #3 (Remarks to the Author):

In the work reported in the manuscript, the authors characterized the structures of microtubules polymerized using S. Pombe tubulins, with and without Mal present. Using S Pombe kinesin motor domains as lattice markers, they confirmed the B lattice structure for S Pombe microtubule by cryo-electron tomography. Based on moiré pattern analysis, they examined the population distribution of S pombe microtubules of various protofilament numbers, as well as the

axis screw variations under the influence of Mal3. Some of these results are unexpected. For example, the 13PF S pombe microtubules show a significant axis-screw, different from that of mammalian microtubules. Also, Mal3 did not show an obvious plus-end binding behavior, and co-polymerization of yeast tubulin and Mal3 did not provide a A lattice. These surprising results are supported by good-quality data in the manuscript. Furthermore, the structural features in the 3D density map agree well with the claimed resolution derived from the FSC. There is still room to improve the Results and Discussion sections to make them more clear and concise. After all, there is rich new and important information in the manuscript that should be made available to the community of cytoskeleton research. So I recommend to publish it.

Responses to Reviewers:

Reviewer #1 (Remarks to the Author):

The authors present a very detailed, structural cryo-electron microscopy (cryo-EM) 3D analysis of S. pombe tubulin and microtubules decorated with the S. pombe Mal3p end-binding protein, a member of the BB1 family. This is the first study of this kind to combine tubulin and Mal3p from their genuine organism (S. pombe). This is a high-resolution study that provides crucial new insights into the function of Mal3p and its interaction with tubulin, and especially near-atomic detail of the binding surface. In a way, this is a continuation of the lab's work on that subject (see: Maurer et al., 2012) that will add additional views and interpretation regarding the mode of action in microtubule end-binding proteins in general.

This is a good piece of work and deserves publication in Nature Communications. The results are very interesting and relevant to a broad scientific audience. Hence, I would recommend publication after some point outlined below have been addressed by the authors.

1. If I'm not mistaken, the Mal3p used here is a C-terminal truncation (as in the Maurer-2012 paper), while others (e.g. Sandblad et al., 2007) worked with a full-length construct. It might be appropriate to acknowledge the, at least SLIGHT possibility that A-lattice recognition, and the results obtained here (and in Maurer et al. 2012) maybe different because of the nature of constructs used. That could be an important issue for function of the native protein, and in vivo and should not be overlooked.

Thank you for highlighting this point. The reviewer is absolutely correct about the different constructs used in each study, and we have now included text concerning this mechanistic possibility in the Discussion (p. 11).

*2. The figure legend of figure 1 is confusing, and probably incomplete:
Figures 1A and 1B look suspiciously like a decoration of microtubules with kinesin motor*

domains ... all of them. How comes? The binding geometry is very different from the Maurer-2012 paper, and from figure 5, but also, the observed mass of the decorating particles look much larger (i.e. like kinesins) than what one would expect from Mal3p. Mal3p has been found to bridge laterally between adjacent protofilaments, hence their sensitivity for the lattice. The densities shown here do not bridge, but stand out exactly as kinesins do.

Thanks for highlighting this potential source of confusion. As is stated in the first paragraph (p5, line 4) of the results section, the legend to Table 1, the Methods section and now additionally in the legend to Fig. 1a, a kinesin motor domain construct was indeed bound to these MTs to allow unambiguous characterization of the underlying lattice architecture in the cryo-tomography experiments depicted in Fig. 1. This was especially important for the differentiation of A- vs B-lattices because - as noted by the reviewer - of the prominence of the motor domain on the outer surface of the MT wall (Fig. 1d, right). As part of this experiment, we confirmed that the presence of the kinesin motor domain did not perturb the overall architecture of a given population of MTs (Supplementary Table 1). Kinesin motor domains were **not** present in the Sp_tub-Mal3-143 sample used for the subsequent single particle data collection and near-atomic resolution structure determination depicted in the remaining figures. This is now also stated in the relevant paragraph of the Results text (p.6).

3. Since the panels are arranged in columns, how are A and B images related? Figure 1A GMPCPP shows two examples, the left microtubule is a 12-pf or 14-pf tube due to the left-right symmetry (has to be an even-pf number), the right one is probably a 15-pf tube, or something else with an odd number of protofilaments. Is this described anywhere, and what are the conclusions? The figure reads as if the authors claim these to be exclusively 13-pf tubes.

As the reviewer points out, and as is indeed shown in Fig. 1c, there are a range of PF architectures in each MT population. Given the unusual architecture of these Sp_tub MTs arising from their PF skew, we considered it important to more directly determine the PF number using the cross-sectional averaging analysis of the 3D tomographic reconstructions depicted in Fig. 1b, as opposed to PF number assignment based on moiré pattern alone. Therefore, we intentionally did not include PF assignment to the 2D MT projections shown in Fig. 1a. The columns of moiré pattern images (Fig. 1a) and the average cross section images (Fig. 1b) are related in that they are derived from the same MT population polymerised under the same conditions indicated in colour at the top – e.g. Fig. 1a,b far left panels are taken from the GMPCPP-Sp_tub population. We have added additional clarification in the legend of Fig. 1b and additional labels concerning the conditions of polymerisation for a and b panels to help differentiate these facets of the data. We present clear examples of the moiré repeats and cross-sections from each population but, given the noisy nature of these tomographic data, these are not necessarily derived from the same MT polymer. This is now stated in the legend to Fig. 1.

4. Figure 2: The result is stunning, but the presentation is not ... For such complex structures the authors should either provide stereo-pairs (works also on screens), or a movie. Figure 2A is too small. Same criticism also applies to figure 3A, B. The differences between sp and mam-tubulin are important, but relatively small and should be better emphasized for the not so frequent viewer of such structures, especially in a journal with a broad audience.

Thanks a lot for the appreciation of the quality of the structure and for the feedback concerning optimising our presentation. We have created a standalone Fig. 2 from panels from the former Fig2. a and b, allowing us to enlarge these panels (renumbering the rest of the figures accordingly), modified former Fig. 3 (new Fig. 4) to include zoom panels of the structural differences between the Sp_tub and Mam_tub models, and have also created a movie of the high resolution reconstruction (Movie S1).

5. Figure 6B-E: With the exception of E, I'm not sure why the authors see a particular preference for tips ... Looks to me as if the green intensity is everywhere, except at the seeds.

Thanks for this feedback – we have rewritten the relevant Results text and figure legend to clarify this point. In summary, the Reviewer has the correct impression because Mal3 rarely shows a preference for MT tips except under very specific conditions (Fig. 7).

6. Are these three different experiments each? How is the frequent bleaching and sudden recovery of rhodamine explained?

In each of panel b-e, each exemplar kymograph represents a different set of conditions, indicated at the bottom of each panel.

The question about rhodamine relates to the changes in fluorescence intensity in the seed – the tubulin in these seeds was actually labelled with Alexa-488 and coloured purple in these views. This is correctly stated in the Methods but incorrectly stated in the previous draft of the legend to this figure – we apologise for this confusion. The variations in fluorescence intensity are due to the length of the movies – typically ~1 hour – during which there are unavoidable intensity variations in the images. Together with occasional data dumps from the microscope where the camera pauses for a couple of frames, these practical aspects of data collection contribute to the appearance of the varying fluorescence from the seed, and are not due to bleaching.

7. The general discussion about A and B lattices is a little extensive ... I believe, these facts are all well known by now. Could be streamlined

According to this suggestion and that of Reviewer 2, we have substantially edited the Introduction, which previously included a general discussion about A and B lattices – we assume this is what was meant by this comment. Text concerning A and B lattices in the Discussion relates specifically to our observations and their context in the literature, so we have not edited this text.

8. I might be wrong, but I believe the convention in the yeast field is to write proteins like Mal3 as Mal3p ...

Obviously, the reviewer is correct that the terminology Mal3p is classically used in yeast genetics to distinguish the protein Mal3p from the gene Mal3. Latterly, however, researchers studying the biophysics and structures of this family of proteins have used the simplified Mal3 nomenclature (Bieling et al (2007); des Georges et al (2008); Katsuki et al (2009); Maurer et al (2011); Maurer et al (2012); Iimori et al (2012); Katsuki et al (2014)), so for consistency with this more recent literature, we would prefer to also simply use “Mal3”. However, we would also welcome editorial guidance on this point.

Reviewer #2 (Remarks to the Author):

The paper from the Moores lab showed a structure of Mal3-bound S. pombe microtubule with at 4.6 Å by cryo-EM with cryo-ET data showing the predominant B-lattice. By performing structural analysis using cryo-electron microscopy (cryo-EM) and in vitro TIRF microscopy assay. The authors showed mainly three things: 1. Mal3 binds to B-lattice of microtubule; 2. Yeast microtubule tubulin lattice does not undergo compaction upon GTP hydrolysis unlike mammalian microtubule; 3. Difference of binding patterns of yeast Mal3 to yeast microtubule from EB1 to mammalian microtubule.

The resolution value authors claim (4.6 Å) seems reasonable to me. The evaluation of the

microtubule structure (compaction and skew) were performed properly.

1. For the first point (EB protein's binding to B-lattice), authors claim that there has been a discrepancy in the field that EB proteins bind to A-lattice or B-lattice microtubule. However, the previous reports suggested that EB proteins bind to the A-lattice are either based on low-resolution metal shadowing (Sandblad et al., 2006), structural analysis without using 3D reconstitution (Katsuki et al., 2014) or low resolution reconstructed structure (~20 Å) (des Georges et al., 2008). Other higher resolution cryo-EM structures all agree that EB proteins bind to B-lattice microtubule except the seam (Maurer et al., 2012; Zhang et al., 2015). In addition, a paper published very recently in JCB (Howes et al., 2017), which is not cited in this manuscript, also showed that EB protein binds to B-lattice microtubule. Thus, the result showing Mal3 binds to B-lattice microtubule is not so surprising.

It is certainly true that our work stands together with recent higher resolution reconstructions of MT-bound Mal3/EB monomers (Maurer et al, 2012; Zhang et al, 2015) in describing Mal3 binding to the MT B-lattice every 8nm. However, the previous, lower resolution work – especially that of des Georges et al (2008) – is still discussed within the literature, so it is a little disingenuous to say that our structural observations are unsurprising. As highlighted by this Reviewer, the work by Howes et al – published at the time of submission of our own manuscript – is obviously relevant to our work, and is discussed in more detail below. In fact, one of the interesting finding from Howes et al - that Bim1 can bind every 4nm within the *S. cerevisiae* tubulin lattice – suggests that Bim1 may be a special case among EBs. In addition, based on the ~1nm resolution MT-Bim1 reconstruction presented by Howes et al, it seems that it cannot be excluded (although extremely unlikely) that Bim1 facilitates formation and stabilisation of A-lattice MTs. In line with the recommendation of Reviewer 1 (comment 1), we have added sentences highlighting ongoing paths of investigation into these discrepancies in the Discussion. Suffice to say, the molecular mechanism(s) of MT binding by these proteins are by no means finalised.

2. In addition, Howes et al. also showed that yeast tubulin lattice does not show compaction as apparent as mammalian tubulin lattice, which compromises significance of the second point of this manuscript. They also discuss about the skew of yeast microtubule lattice. The only difference is that Howes et al. used EB protein Bim1 and tubulin derived from budding yeast instead of fission yeast as in this manuscript.

The work by Howes et al is obviously highly relevant to our discussion of the different properties of tubulins from yeasts and mammals. As we now describe below and in new text in the Discussion, there are both similarities and numerous interesting differences between the properties and interaction of *S. cerevisiae* tubulin (Sc_tub) and its EB (Bim1) described by Howes et al, and our findings concerning the Mal3+Sp_tub complex. This comparison raises intriguing questions about the conservation or lack of it amongst MT-EB interactions, further emphasising that the molecular mechanisms of this family of proteins are by no means finalised, thereby providing ample justification for the timeliness and relevance of our study to the field.

- Similarity: Both studies investigate the MT binding properties of an EB family protein, either Bim1 (Sc_tub) or Mal3 (Sp_tub), to the relevant yeast tubulin.
- Difference: Whereas our Mal3-143 construct terminates within the flexible linker region between the MT-binding CH domain and the dimerising coiled coil, the Bim1-EGFP (1-210) includes some of the dimerisation region of this protein, as well as an EGFP tag, which itself is prone to dimerisation. Another difference in sample preparation is that we did not use a nucleotide analogue in our single particle reconstruction of Mal3+Sp_tub at 4.6 Å resolution, whereas Howes et al used GTP γ S to determine the structure of Bim1+Sc_tub at ~ 10 Å resolution. Although no specific information is provided by Howes et al concerning the resolution of the Sc_tub-Bim1 reconstruction, comparison of the definition of the CH domain

fold in our Fig. 5A and Howes et al Fig. 3K, suggests an overall resolution in their reconstruction of ~1nm in the tubulin with slightly worse resolution in the Bim1 density. They describe the challenges of working with their samples without this nucleotide analogue, which we did not experience, further highlighting intrinsic differences between the tubulins/EBs from these two sources.

- Similarity: In providing the lattice parameters for different Sc_tub MTs derived during single particle reconstruction, the presence of PF skew can be inferred in these MTs, as we characterised by direct observation using cryo-electron tomography of Sp_tub MTs and direct measurement of their moiré repeats (Fig. 1, new Supplementary Fig. 4). In both systems, cognate EB protein binding appears to slightly modify PF skew.

- Difference: Whereas the PF skew in Sc_tub MTs is only moderately sensitive to tubulin-bound nucleotide, we show that the skew in GTP-vs GMPCPP-bound Sp_tub is extremely different (Fig. 1c).

- Similarity: Both studies used yeast tubulin purified from the relevant yeast species.

- Difference: Sc_tub used by Howes et al has a His-tag (explicitly described in Johnson et al (2011) Biochemistry) and is over expressed at least in some samples, whereas Sp_tub used in our study is untagged and expressed using a native promoter (Drummond et al, 2011).

- Similarity: Neither Sc_tub nor Sp_tub dynamic MTs exhibit GTPase-dependent compaction

- Difference: Bim1 binding does cause compaction of Sc_tub, whereas Mal3 binding does not cause compaction of Sp_tub. Howes et al identify possible Sc_tub specific substitutions compared to Mam_tub that may be responsible for the lack of structural compaction in dynamic Sc_tub MTs: G98S (β -tubulin) and T254N (α -tubulin). This substitution is not seen in Sp_ β -tubulin (Supplementary Fig. 3) whereas in Sp_ α -tubulin, the equivalent residue is N257 in Sp_ α -tubulin1 and A253 in Sp_ α -tubulin2 (Supplementary Fig. 2). As depicted in Fig. 3a in our revised manuscript, non-conserved residues are distributed across the Sp_tub dimer structure and we anticipate that they collectively contribute to species-specific differences in tubulin mechanochemistry.

- Similarity: Both Bim1 and Mal3 bind between PFs in the MT wall.

- Difference: As already described above, Howes et al describe the 4nm MT binding pattern for Bim1, whereas the 8nm binding pattern for Mal3 is a very clear feature of our data (Supplementary Fig. 1a) even while working at higher Mal3 concentrations than were used for Bim1. The 8nm binding pattern of Mal3 allows computational differentiation between α -tubulin and β -tubulin enabling calculation of our near-atomic resolution reconstruction. The 4nm binding pattern for Bim1 is a very surprising finding and we speculate that this could be an Sc_tub specific property. Howes et al propose specific residues that may be the origin of this property in Bim1 compared to mammalian EB, one of which is also present in Mal3 (H69Y). As with differences with the properties of the tubulins, sequence differences throughout the EB structure likely contribute to differences in the behaviours of these proteins.

3. For the third point, authors used PISA and homology modeling to claim that Mal3 has evolved such that it is favorable for binding to yeast extended microtubule lattice. Rather, authors should include the result of binding pattern of S. cerevisiae EB protein Bim1 to microtubule reported in Howes et al. (2017) and compare with their structure to draw more accurate comparison.

The model coordinates derived from the Sc_tub-Bim1 reconstruction – which are essential to perform the suggested PISA analysis for this structure - are not deposited in the public structural databases. Howes et al performed homology modelling based on the high resolution structure of Mam_tub-EB3, which we have already included in our current analysis (Supplementary Fig. 6b). In addition, PISA is designed to analyse macromolecular

interfaces within higher resolution structures and is likely to be susceptible to uncertainties intrinsic to calculation of the Bim1 homology model from a lower resolution reconstruction. Nevertheless, to accompany our discussion of Bim1+Sc_tub structure with Mal3+Sp_tub, we have now also included the Bim1 sequence in the sequence alignment in Supplementary Fig. 6a (now 7a), from which the differences that may contribute to their different MT binding properties can be considered.

*4. For the part of TIRFM assay, it is really hard to make sense the results since tubulin concentrations are different in each experiment (4.5 μ M for *S. pombe* tubulin in Fig.6b and c, 25 μ M for brain tubulin in Fig.6d and 14 μ M for *S. pombe* tubulin in Fig.6e) and KCl concentrations are not consecutive (0-55 mM KCl for Mal3 vs brain tubulin and 55-85 mM KCl for Mal3 vs *S. pombe* tubulin).*

Thanks for this feedback – we have rewritten the relevant Results text and figure legend to clarify these potential sources of confusion.

5. The authors' most important claim here is that Mal3 is less able to differentiate between GTP-rich tip and GDP-rich lattice of yeast microtubule since both have similar structures. However, to me, the results appear completely opposite. In Fig.6e, Mal3 is clearly localized at the tip at 55 mM and 75 mM KCl conditions. The difference in Mal3 intensities of tip and lattice appears more prominent compared with the brain tubulin result in Fig.6d which can undergo compaction. If authors are to convince the readers, authors have to; 1, show clearer kymograph (the first kymograph in Fig.6b and e are hard to see since the growing microtubule are too short) and 2, present quantitative data comparing intensities of GFP at the tip and the lattice.

Thank you for this feedback. Our point here is that for *S. pombe* MTs but not for mammalian MTs, we needed to tune buffer conditions and protein concentrations to an extraordinary extent in order to produce "tip tracking" by Mal3.

Fig. 7 b,c (old Fig. 6) show that under "standard" conditions (PIPES, EDTA, Mg, as commonly used for MT dynamics experiments), neither EB1 nor Mal3 "sees" a difference between the tip and the lattice of dynamic *S. pombe* MTs. Panel d shows in contrast that for brain MTs under these standard conditions, Mal3 DOES see a difference between tip and lattice. Panel e shows that by going to extraordinary conditions of 14 μ M tubulin and 55-85 mM KCl, preferential binding of Mal3 to the tips of *S. pombe* MTs can, ultimately, be visualised. The message is that Mal3 can see a structural difference between tip and lattice of *S. pombe* MTs, but the difference is slight, so that we need to drive the system to unusual conditions in order to detect it. The MTs are short in these kymographs (panel e) because the conditions are so extreme. Otherwise we saw MTs but no difference between tip and lattice binding, as shown in panels b and c.

The requirement to tune solution conditions in order to increase the "contrast" between tip and lattice binding is something that is not commonly emphasised in tip tracking papers and we are trying here to present a more comprehensive, realistic but qualitative view. Given the required tuning, undertaking quantitative data analysis on non-equivalent assays conditions seems to us not meaningful. We have rewritten the relevant Results text and figure legend to help clarify this point further.

6. Also, the writing of the current manuscript is not suitable for a general audience. The author has very MT specific introduction, which is hard to read. Therefore, I feel like this paper is either not suitable for publication in Nature Communications or it needs a major revision to differentiate its finding by comparing their structure with structures reported in Howes et al., 2017.

Following this suggestion and that of Reviewer 1, we have substantially streamlined the Introduction text, edited throughout the Results text and, as outlined above, have now included new text in the Discussion concerning the comparison with the highly relevant work of Howes et al.

Minor points:

7. In the Introduction, authors are citing ref 20, 23 and 24 as examples of “higher resolution cryo-EM studies show EB binding to the B-lattice”, but I would not consider citation 24 (Guesdon et al., 2016) as high-resolution structure showing EB binding pattern.

Thanks for pointing this imprecision out – we have now adjusted this text accordingly.

8. Authors claimed that docking of EB3+Mam_tub E-site structure is a good match with their Mal3-143+Sp_tubu lattice density map, but it is difficult to see from the figure. Different views of Fig. 3c, d and e as Supplementary Figures would be necessary. Also, In Fig.3d, there is a green letter “α” on top of GDP molecule. I think this is a mistake happened when authors were preparing the figure.

Thanks for this suggestion – we have included these additional views for comparison in a new Supplementary Fig. 5. Apologies for the stray “α” in Fig. 3d and thanks for pointing this out – we have removed it.

9. In the cryo-EM part in the RESULTS, authors are saying that there are two isoforms of α-tubulin as “reconstruction uses a wild-type mixture of Sp_α-tubulin1/2 isoforms”. However, in the section of TIRF microscopy of Mal3-GFP and Eb1-GFP binding to dynamic MTs in vitro of METHODS, authors are saying that “unlabeled single isoform S. pombe tubulin” were used. I could not find mentions about purifying single isoform S. pombe tubulin in the METHODS, but what is the difference?

We have now added relevant information to the Methods text (p14).

Other minor issues:

*10. For Fig.3f legend, I think it’s common to write like “red asterisk” instead of writing like “red *”. Same with the legend of Fig.4 and legend of Supplementary Fig.5. Also for third paragraph (Fig. 3f, red*) and fourth paragraph (Fig4a, *) in p.7.*

Thanks for pointing this out – we have now fixed the text in the places mentioned.

11. Citation format is different only at the last part of the second paragraph in p.5. “and are consistent with the PFs in Sp_tub MTs being markedly more skewed within the lattice (19).”

Thanks for pointing this out – we have now fixed that formatting error.

12. There are places without space between numbers and units throughout the manuscript.

Thanks for pointing this out – we have now fixed these formatting errors.

13. In the section of Cryo-EM grid preparation and data collection in the METHODS, it is “holey carbon grids”, not “holy carbon grids”.

Thanks for pointing this out – we have now fixed this text.

Reviewer #3 (Remarks to the Author):

In the work reported in the manuscript, the authors characterized the structures of microtubules polymerized using S. Pombe tubulins, with and without Mal present. Using S Pombe kinesin motor domains as lattice markers, they confirmed the B lattice structure for S Pombe microtubule by cryo-electron tomography. Based on moiré pattern analysis, they examined the population distribution of S pombe microtubules of various protofilament numbers, as well as the axis screw variations under the influence of Mal3. Some of these results are unexpected. For example, the 13PF S pombe microtubules show a significant axis-screw, different from that of mammalian microtubules. Also, Mal3 did not show an obvious plus-end binding behavior, and co-polymerization of yeast tubulin and Mal3 did not provide a A lattice. These surprising results are supported by good-quality data in the manuscript. Furthermore, the structural features in the 3D density map agree well with the claimed resolution derived from the FSC.

There is still room to improve the Results and Discussion sections to make them more clear and concise.

We thank the Reviewer for their positive view of our work. According to their suggestion and those of the other Reviewers, we have edited our text extensively throughout, aiming to make it more clear and concise.

After all, there is rich new and important information in the manuscript that should be made available to the community of cytoskeleton research. So I recommend to publish it.

Reviewers' Comments:

Reviewer #1 (Remarks to the Author):

Overall, the author did a great job addressing my concerns. I was already pleased with this work the firsts round, but now I believe this is ready to go. My only remaining question relates to something I obviously overlooked the first time, and relates to Figure 1: When co-decorating Mal3p with KMD's, is the Mal3p binding not affected, e.g. by steric hinderance? How could the authors tell. Or, am I getting this wrong? I assume, the authors rely on a preserved Mal3 modulated microtubule lattice that survives post-decoration with KMD?

The addition of the movie is very helpful, and the figures have been improved adequately. Hence, I recommend publication.

Reviewer #2 (Remarks to the Author):

I really appreciate that the authors have made tremendous efforts to improve the manuscript and address the reviewers' concerns and suggestions. The readability of the manuscript is much better now instead of being too microtubule specific. The figure is improved greatly. The manuscript now contains a segment in the discussion about the recent structure of *S. cerevisiae* MT. I think this part is adequately addressed the similarity and differences between the two studies and highlight the intriguing questions about the conservation of EB1 and effects of tubulin specific sequence to the microtubule structure.

At this point, I think the manuscript is appropriate for publications in Nature Communications.

Minor points:

1) Supplementary Figure S5 was not mentioned anywhere in the text.

2) There is one small thing that I would suggest the author to try, which may help to differentiate the nucleotide occupancy of the N-sites and E-sites.

The author can use the atomic model of the α -tubulin and the β -tubulin to simulate a map at 4.6Angstrom resolution (Chimera molmap function), then calculate the difference map between their map and the simulated map. The difference map will be the density for the nucleotide

Responses to Reviewers:

Reviewer #1 (Remarks to the Author):

Overall, the author did a great job addressing my concerns. I was already pleased with this work the firsts round, but now I believe this is ready to go. My only remaining question relates to something I obviously overlooked the first time, and relates to Figure1: When co-decorating Mal3p with KMD's, is the Mal3p binding not affected, e.g. by steric hinderance? How could the authors tell. Or, am I getting this wrong? I assume, the authors rely on a preserved Mal3 modulated microtubule lattice that survives post-decoration with KMD?

Thanks for the encouraging comments and great help in improving our manuscript. The cryo-electron tomography (cryo-ET) experiments depicted in Fig. 1 (see also Table 1) were conducted to establish a) the underlying architecture of Sp_tub MTs and b) the effect on this architecture of Mal3. The presence of the KMD fiducial marker allowed the clearest depiction of the lattice and was especially important for visualising the B-lattice contacts. To be sure that the KMD itself did not perturb the MT architecture – the possibility the Reviewer highlights here - we also performed a subset of these experiments in the absence of KMD (Supplementary Table 1). Since the overall lattice parameters measured \pm KMD are essentially the same in each condition, we concluded that the KMD marker did not itself perturb the overall MT architecture, at least at the resolution captured in this cryo-ET analysis.

Reviewer #2 (Remarks to the Author):

I really appreciate that the authors have made tremendous efforts to improve the manuscript and address the reviewers' concerns and suggestions. The readability of the manuscript is much better now instead of being too microtubule specific. The figure is improved greatly. The manuscript now contains a segment in the discussion about the recent structure of S. cerevisiae MT. I think this part is adequately addressed the similarity and differences between the two studies and highlight the intriguing questions about the conservation of EB1 and effects of tubulin specific sequence to the microtubule structure.

At this point, I think the manuscript is appropriate for publications in Nature Communications.

Thanks for the encouraging comments and great help in improving our manuscript.

Minor points:

1) Supplementary Figure S5 was not mentioned anywhere in the text.

Thanks for pointing this out and sorry for missing it – this figure is now referenced on p7.

2) There is one small thing that I would suggest the author to try, which may help to differentiate the nucleotide occupancy of the N-sites and E-sites. The author can use the atomic model of the α -tubulin and the β -tubulin to simulate a map at 4.6Angstrom resolution (Chimera molmap function), then calculate the difference map between their map and the simulated map. The difference map will be the density for the nucleotide

Thanks a lot for this nice suggestion. We have included these density views in Supplementary Fig. 5 and reference this suggestive density in the main text (p7), which supports our conclusions.